# Decision-aware Training of Spatiotemporal Forecasting Models to Select a Top-K Subset of Sites for Intervention

**Kyle Heuton** [1]   **F. Samuel Muench** [1]   **Shikhar Shrestha** [2]   **Thomas J. Stopka** [2]   **Michael C. Hughes** [1]

## Abstract

Optimal allocation of scarce resources is a common problem for decision makers faced with choosing a limited number of locations for intervention. Spatiotemporal prediction models could make such decisions data-driven. A recent performance metric called fraction of best possible reach (BPR) measures the impact of using a model's recommended size K subset of sites compared to the best possible top-K in hindsight. We tackle two open problems related to BPR. First, we explore *how to rank* all sites numerically given a probabilistic model that predicts event counts jointly across sites. Ranking via the per-site mean is suboptimal for BPR. Instead, we offer a better ranking for BPR backed by decision theory. Second, we explore *how to train* a probabilistic model's parameters to maximize BPR. Discrete selection of K sites implies all-zero parameter gradients which prevent standard gradient training. We overcome this barrier via advances in perturbed optimizers. We further suggest a training objective that combines likelihood with a BPR constraint to deliver high-quality top-K rankings as well as good forecasts for all sites. We demonstrate our approach on two where-to-intervene applications: mitigating opioid-related fatal overdoses for public health and monitoring endangered wildlife.

## 1. Introduction

Statistical machine learning methods for spatiotemporal forecasting can play a vital role in high-stakes applications from mitigating overdoses in public health (Marks et al., 2021b) to forest fire management (Cheng & Wang, 2008) to wildlife monitoring (Golden et al., 2022; Hefley et al., 2017). Across these domains, there is a pressing need for predictive models that can make accurate predictions of near-term future events at fine spatiotemporal resolutions. Such models can enable inform data-driven decisions about how to allocate limited resources to maximize utility.

In this work, we seek to help decision-makers select *where to intervene*. Given historical data for a fixed set of $S$ candidate spatial sites, we develop models that can recommend a specific subset of given size $K$ for some action or intervention. We think of hyperparameter $K$ as setting the budget for interventions. In an ideal world, decision-makers could afford interventions in all $S$ sites. However, when resource constraints allow only $K$ sites to receive interventions, a decision selecting a specific $K$-of-$S$ subset is required. While such decisions may often be heuristic in current practice, we hope to offer data-driven solutions.

With this goal in mind, choosing a sensible performance metric is critical to assessing which models have real-world utility. Common metrics such as squared error or absolute error are not well matched to where-to-intervene decisions because they treat all sites equally. Recent work on overdose forecasting has suggested a metric termed the *fraction of best possible reach*, or *BPR* (Heuton et al., 2022; 2024). BPR measures a ratio of event counts. The numerator sums over the model's recommended $K$ sites, while the denominator sums over the best possible $K$ selected in hindsight. This type of evaluation has been used in a preregistered trial (Marshall et al., 2022) for assessing forecasts of opioid overdoses in Rhode Island, as well as a follow-up feasability study (Allen et al., 2023). BPR is applicable to many where-to-intervene problems beyond public health.

While some publications have reported BPR in evaluations, we suggest that a natural goal would be for this performance metric to inform two other key parts of data-driven decision-making: model-based *ranking* and model *training*. By ranking, we mean that given fixed model parameters, the knowledge of BPR as the metric of interest should im-

---

[1]Department of Computer Science, Tufts University, Medford, Massachusetts, United States [2] Department of Public Health and Community Medicine, Tufts University School of Medicine, Boston, Massachusetts, United States. Correspondence to: Kyle Heuton <kyle.heuton@tufts.edu>.

*Proceedings of the 42nd International Conference on Machine Learning*, Vancouver, Canada. PMLR 267, 2025. Copyright 2025 by the author(s).

---

Code: https://github.com/tufts-ml/decision-aware-topk

pact the numerical score assigned to each site to determine the top $K$. By training, we mean how to update model parameters to achieve high BPR. This paper contributes new methods for solving both ranking and training problems when BPR is the preferred metric.

Our work overcomes several technical barriers. The first barrier is in ranking. Given a fixed probabilistic model, determining how to compute a per-site score, which will later be sorted to find the top $K$, is not obvious. It may be tempting to use the model's per-site mean, but decision theory suggests not all loss functions recommend the per-site mean as the best estimator (Berger, 2013; Murphy, 2022). As a relatively new metric, the problem of how to assign a numerical ranking to sites for optimal BPR decision-making is currently open. We contribute a tractable ranking method that is provably best for a reasonable bound on BPR. We further show how the *score function estimator* (Kleijnen & Rubinstein, 1996; Mohamed et al., 2020) can be used to calculate gradients of this ranking with respect to parameters.

The second barrier prevents training parameters to improve BPR. First-order gradient descent is a common, effective algorithm we would like to use. However, gradients of BPR with respect to parameters are problematic. While small changes to parameters induce some changes to per-site scores, only changes large enough to move a site into or out of the top-K ranked sites will adjust BPR. Thus, gradients of BPR with respect to parameters will be zero almost everywhere, preventing gradient methods from ever moving beyond subpar initial parameters. To fix this, we leverage recent advances in *perturbed optimizers* (Abernethy et al., 2016; Berthet et al., 2020) to yield effective and efficient gradient estimation for BPR. Our team explored this idea for optimizing BPR alone in earlier non-archival work (Heuton et al., 2023); this paper offers an expanded treatment with more accessible presentation, while addressing two more barriers.

The final barrier is designing a training objective to achieve applied goals. We find that optimizing BPR alone can lead to predictions with far lower likelihood than conventional training. This raises concerns about overall model quality and generalization. To address this, we pose a constrained optimization problem to maximize likelihood subject to a BPR quality constraint. This combined objective delivers quality top-K recommendations *and* good forecasts for all sites. Our objective is reminiscent of past additive combinations of a regression loss and decision loss (Kao et al., 2009). Unlike that work, ours pursues non-convex losses and directly enforces decision quality via constraints.

We ultimately contribute methods for how-to-rank and how-to-train when making where-to-intervene decisions. Using these tools, a variety of models can be directly optimized to make effective top $K$ site recommendations. We demon-strate these contributions first on synthetic data, where we reveal how off-the-shelf methods without our innovations can be suboptimal for decision-making. We further evaluate against alternatives on two applications: mitigating opioid-related fatal overdoses and monitoring endangered birds. We hope our contributions spark interest in where-to-intervene problems in the methodological community and also lead to effective deployments of data-driven top-K decision-making in public health and beyond.

**Related work.** The application of machine learning to decision-making problems is widely studied in operations research literature (Bertsimas & Kallus, 2020; Sadana et al., 2025), including the problem of how to train a model for downstream decision-making (Mandi et al., 2024). We review several model training approaches later in Sec. 4.

Other researchers have used decision-aware objectives to solve limited resource allocation problems. Chung et al. (2022) study how to allocate essential medicines in Sierra Leone across hospital sites. Gupta et al. (2024) pursue a where-to-intervene task in urban planning, selecting where to build speed humps to reduce pedestrian injuries. Our work differs in its focus on the BPR metric, our hybrid decision-aware objective that preserves likelihood, and evaluations that forecast the *future* given the recent past.

Throughout this paper, a recurring takeaway is that conventional training based on maximizing likelihood can yield suboptimal decision-making for BPR, especially when forecasting models are misspecified. In this vein, our work shares similar goals as *direct loss minimization* (Wei et al., 2021) and *loss-calibrated* methods (Lacoste–Julien et al., 2011). We are inspired by the way these works combine task-specific losses and decision theory to improve probabilistic models. Others have extended loss-calibration to neural nets (Cobb et al., 2018) and to continuous actions (Kuśmier-czyk et al., 2019). Yet a tractable and scalable recipe that prioritizes top-$K$ where-to-intervene decision-making is not an immediate next step from this work.

## 2. Technical Background and Problem Setup

**Notation.** Mathematically, some of our notation follows Sander et al. (2023). Function TOPKMASK takes as input a vector $\boldsymbol{r}$ of length $S$ and an integer $K$. It produces a binary vector of length $S$ with exactly $K$ entries equal to 1. Each 1 entry corresponds to a value in the top $K$ largest entries of the input vector. The $s$-th entry of the output is

$$\text{TOPKMASK}(\boldsymbol{r}, K)_s = \begin{cases} 1 & \text{if } \text{RANK}(\boldsymbol{r})_s \leq K \\ 0 & \text{otherwise} \end{cases}, \quad (1)$$

Here, function RANK provides a numerical ranking (largest-to-smallest) from 1 to $S$ for each entry of an $S$-dimensional input vector. The function TOPKIDS acts on the same input

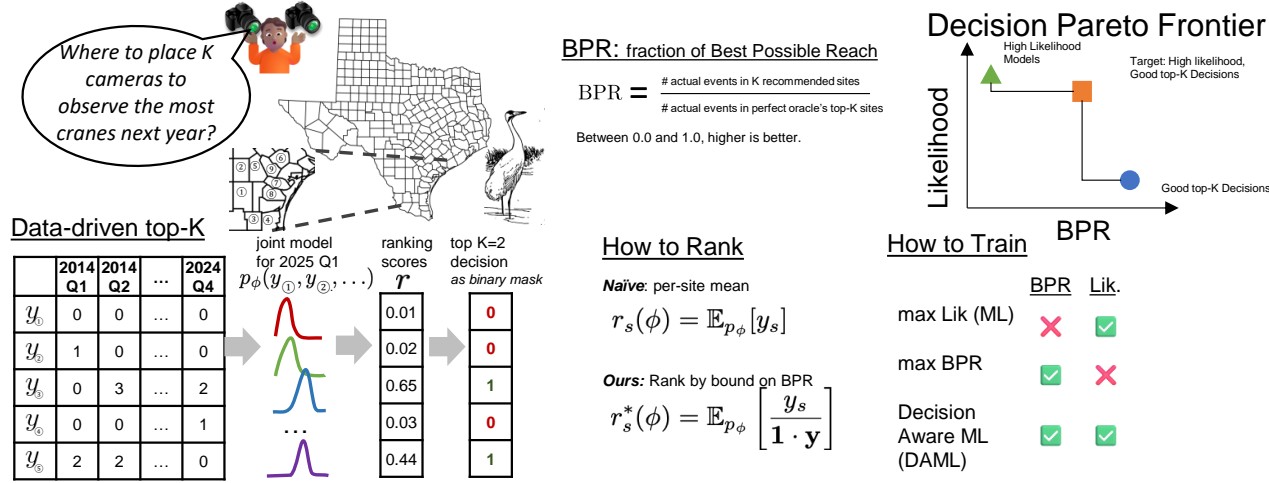

Figure 1: Visual overview of our approach and contributions to the *how to rank* and *how to train* open problems.

as TOPKMASK, but return the size $K$ set of integer indices corresponding to top $K$ values.

**Problem definition.** We wish to probabilistically model events that occur across $S$ distinct spatial sites over time. At each site, indexed by $s$, we can observe a *non-negative* scalar count or value $y_s \geq 0$. We assume that larger $y_s$ corresponds to greater value in intervention at site $s$. In our public health applications, $y_s$ represents counts of fatal opioid-related overdoses. In our later wildlife monitoring case study, $y_s$ counts how often a rare animal appears. At each time $t$, we stack all observations into a vector $\boldsymbol{y}_t \in \mathbb{R}^S$. We assume that the true data-generating distribution for each vector $\boldsymbol{y}_t$ given past history does not change over time, including between the training and test periods.

We denote our joint model for this vector as $p_\phi(\boldsymbol{y}_t)$, where model parameter vector $\phi$ defines the density over r.v. $\boldsymbol{y}_t$. We assume that explicitly evaluating this pdf and sampling values of $\boldsymbol{y}_t$ are both feasible. Given these assumptions, our framework is quite flexible: the vector $\phi$ could represent the weights of a neural network or the coefficients of a logistic regression or a Bayesian hierarchical model.

Given a training set of $T$ times, our model family in general factorizes $p(\boldsymbol{y}_{1:T}) = \prod_{t=1}^{T} p_\phi(\boldsymbol{y}_t | \boldsymbol{y}_{1:t-1})$. To ease notation throughout, we omit conditioning on past history or other exogenous features. So $p_\phi(\boldsymbol{y}_t)$ below should be read as equal to $p_\phi(\boldsymbol{y}_t | \boldsymbol{y}_{1:t-1})$ for models with such dependencies.

Our goal is to use this probabilistic model to solve a where-to-intervene decision making problem. We primarily intend to use the model to numerically rank all $S$ sites, then select the top $K$ sites in this ranking for near-future intervention.

**Definition of BPR.** At current time $t$, we evaluate a model $\phi$'s ability to select a top-$K$ subset of sites for intervention before time $t+1$. Let $\mathcal{R}$ be the model's recommended subset of $K$ sites among all $S$ sites. For the rest of this section, let $\boldsymbol{y} \in \mathbb{R}^S$ be the vector of observations at the target time $t+1$ (we skip time subscripts on $\boldsymbol{y}$ to keep notation simple). The vector $\boldsymbol{y}$ is not available when the decision of $\mathcal{R}$ is made. Following Heuton et al. (2022), we define BPR as:

$$\text{BPR}(\mathcal{R}, \boldsymbol{y}) = \frac{\sum_{s \in \mathcal{R}} y_s}{\sum_{s \in \text{TOPKIDS}(\boldsymbol{y}, K)} y_s}. \quad (2)$$

Both terms in this fraction can only be evaluated in hindsight, after the vector $\boldsymbol{y}$ is realized at time $t+1$. The numerator counts how many events the model's recommendation would reach. The denominator counts how many events a perfect oracle with knowledge of the future could reach on the same budget of $K$ sites. Overall, we interpret BPR as the fraction of events of interest the current model's selection $\mathcal{R}$ would reach compared to perfect knowledge of the future. Higher BPR indicates a better model for choosing where to intervene. BPR's best value is 1.0, its worst is 0.0.

**Ranking sites.** Given a fixed model $\phi$ and target time, the *ranking* problem is how to assign numerical values to all $S$ sites so that if the top $K$ sites are assigned to $\mathcal{R}$, we reap high BPR scores. We need to define a ranking vector $\boldsymbol{r} \in \mathbb{R}^S$ of numerical scalar scores for all $S$ sites. Higher $r_s$ values indicate greater priority for site $s$.

Suppose we have a loss function $L(\boldsymbol{r}, \boldsymbol{y})$ (not necessarily related to BPR) that produces a scalar value indicating the overall quality of taking an "action" $\boldsymbol{r}$ and then realizing outcome $\boldsymbol{y}$. Lower values of $L$ indicate better decisions. A natural framework for making decisions about actions (Murphy, 2022) is to minimize the expected loss:

$$r^* = \operatorname*{argmin}_{\boldsymbol{r}} \mathbb{E}_{\boldsymbol{y} \sim p_\phi} [L(\boldsymbol{r}, \boldsymbol{y})] \quad (3)$$

As a simplistic example, if the loss is defined as the sum of squared errors, $\mathcal{L}(\boldsymbol{r}, \boldsymbol{y}) = \sum_{s=1}^{S} (r_s - y_s)^2$, the optimal ac-

tion is provably the per-site mean: $r_s = \mathbb{E}_{p_\phi}[y_s]$. Similarly, for the loss that sums up *absolute* errors, the optimal action is the per-site *median* (Schwertman et al., 1990; Balkus, 2024). We tackle defining an optimal ranking for BPR.

To pose our ranking problem formally, we need to convert the higher-is-better BPR metric into a lower-is-better loss that depends on $\mathbf{r}$. Define *negative* BPR loss as

$$L^{\mathrm{BPR}}(\mathbf{r}, \mathbf{y}) = -\frac{\mathbf{y} \cdot \mathrm{TOPKMASK}(\mathbf{r}, K)}{\mathbf{y} \cdot \mathrm{TOPKMASK}(\mathbf{y}, K)}. \qquad (4)$$

This way of writing the loss with dot products of top-K binary vectors is equivalent to $-\mathrm{BPR}(\mathrm{TOPKIDS}(\mathbf{r}, K), \mathbf{y})$.

**Connection to 0-1 knapsack.** Given a fixed $\mathbf{y}$ vector, the problem of selecting $K$ sites to minimize $L^{\mathrm{BPR}}$ can reduce to the canonical 0-1 knapsack problem (Dantzig, 1957) where each site $s$ would have value $y_s$ and weight 1, and the budget constraint allows just $K$ of all $S$ sites. Our how-to-rank contribution solves a more general problem: how to set $\mathbf{r}$ when $\mathbf{y}$ is not given but must be forecasted by our model.

In Sec. 3 below, we show how analysis of tractable bounds of the loss in Eq. (4) suggests a high-quality ranking function $\mathbf{r}^*(\phi)$ for BPR. This ranking is usable across different model families, as long as the model $p_\phi$ allows generating many samples of events $\mathbf{y}$. Later in Sec. 4, we show how to *train* parameters $\phi$ with gradient descent to yield better top-K decisions as judged by BPR.

## 3. Methods for Ranking

**Loose bound justifies per-site mean ranking.** A natural first guess for ranking is the per-site mean: $\bar{\mathbf{r}} = \mathbb{E}_{p_\phi}[\mathbf{y}]$. We can show this is justifiable way to minimize expected loss on a simplistic upper bound on BPR. Assume there exists an *upper limit* $U$ such that for all $s$, we can guarantee $U \geq y_s$. The sum over any $K$ entries of vector $\mathbf{y}$ in the denominator of BPR is then bounded by $K \cdot U$. Plugging this bound into the minimize expected loss problem and simplifying with linearity of expectations yields

$$\mathbf{r}^* \leftarrow \underset{\mathbf{r}}{\mathrm{argmin}} - \underbrace{\frac{\mathbb{E}_{p_\phi}[\mathbf{y}]}{K \cdot U} \cdot \mathrm{TOPKMASK}(\mathbf{r}, K)}_{\leq \mathrm{BPR}(\mathbf{r}, \mathbf{y})} \qquad (5)$$

Many solutions exist: any vector $\mathbf{r}^*$ that satisfies $\mathrm{TOPKMASK}(\mathbf{r}^*, K) = \mathrm{TOPKMASK}(\mathbb{E}_{p_\phi}[\mathbf{y}], K)$ can be an argmin. One valid solution here is the per-site mean $\bar{\mathbf{r}}(\phi) = \mathbb{E}_{p_\phi}[\mathbf{y}]$. However, this solution is optimal for a potentially quite loose bound on BPR that approximates the denominator with the constant $K \cdot U$.

**Tighter bound suggests the ratio estimator for ranking.** Instead of bounding with constant $K \cdot U$, we can upper bound the denominator in Eq. (4) by summing over

all $S$ terms instead of the top $K$: $\sum_{s \in \mathrm{TOPKIDS}(\mathbf{y})} y_s \leq \sum_{s=1}^{S} y_s = \mathbf{1} \cdot \mathbf{y}$. This bound is tighter when the sum of all entries is less than $K \cdot U$, which is typically true of sparse $\mathbf{y}$ vectors in our applications. Our how-to-rank problem then becomes

$$\underset{\mathbf{r}}{\mathrm{argmin}} - \underbrace{\mathbb{E}_{\mathbf{y} \sim p_\phi}\left[\frac{\mathbf{y}}{\mathbf{1} \cdot \mathbf{y}}\right] \circ \mathrm{TOPKMASK}(\mathbf{r}, K)}_{\leq \mathrm{BPR}(\mathbf{r}, \mathbf{y})} \quad (6)$$

Again, we solve via any vector $\mathbf{r}^*$ whose top-K binary vector $\mathrm{TOPKMASK}(\mathbf{r}^*, K)$ equals $\mathrm{TOPKMASK}(\mathbb{E}_{p_\phi}[\frac{\mathbf{y}}{\mathbf{1} \cdot \mathbf{y}}], K)$. One valid solution is the expected ratio of vector $\mathbf{y}$ to its sum, which we nickname the *ratio estimator*

$$\mathbf{r}^*(\phi) = \mathbb{E}_{\mathbf{y} \sim p_\phi}\left[\frac{\mathbf{y}}{\mathbf{1} \cdot \mathbf{y}}\right] \approx \frac{1}{M} \sum_{m=1}^{M} \frac{\mathbf{y}^{(m)}}{\mathbf{1} \cdot \mathbf{y}^{(m)}}. \quad (7)$$

This ranking is *distinct* from the per-site mean: there exist fixed models $\phi$ where the ratio estimator and the per-site mean would select different subsets of the same $S$ sites. See App. B for concrete cases where the ratio estimator earns BPR 2.5x to 5x higher than the per-site mean, even when all estimators know the true data-generating model.

When exact computation of this expectation is not easy, we recommend a Monte Carlo approximation using $M$ samples $\{\mathbf{y}^{(m)}\}_{m=1}^{M}$ drawn iid from $p_\phi$, as in Eq. (7). This is a stochastic estimator; rankings can differ across repeat trials if $M$ is not large enough.

## 4. Methods for Training

We now consider various ways to train the parameters of our probabilistic model on a training set that covers $T$ distinct time periods indexed by $t$.

### 4.1. Maximum likelihood (ML) estimation

Conventional training would maximize the likelihood, or equivalently minimize negative log likelihood (NLL):

$$\mathcal{J}^{\mathrm{NLL}}(\phi) = -\sum_{t=1}^{T} \log p_\phi(\mathbf{y}_t). \qquad (8)$$

We assume $p_\phi$ is differentiable, so solving for a point estimate $\phi$ is possible via gradient descent. If we add an optional prior term $\log p(\phi)$ to enforce an inductive bias or control over-fitting, this is known as *MAP* estimation.

If the model is *well-specified* and training set size $T$ is large enough, this is a reliable strategy to estimate $\phi$. We could then use the ranking methods from Sec. 3 for where-to-intervene decisions. However, popular wisdom reminds us that "all models are wrong" in some way for real-world data. As we will show in later experiments, fitting a *misspecified* model via ML estimation can produce $\phi$ that deliver suboptimal BPR, even using the optimal ranking for that $\phi$.

## 4.2. Direct loss minimization for BPR

Inspired by the broad goal of *direct loss minimization* (Wei et al., 2021), another approach would be to find parameters that minimize our BPR-specific decision making loss $L^{\text{BPR}}$. In this strategy, we seek $\phi$ values that minimize

$$\mathcal{J}^{\text{BPR}}(\phi) = \sum_{t=1}^{T} L^{\text{BPR}}(\boldsymbol{r}_t^*(\phi), \boldsymbol{y}_t), \qquad (9)$$

Here, for $\boldsymbol{r}^*(\phi)$ we use the ratio estimator in Eq. (7).

To train with modern gradient methods, we'd need to compute the gradient $\nabla_\phi \mathcal{J} = \sum_t \nabla_\phi \boldsymbol{r}_t \cdot \nabla_{\boldsymbol{r}_t} L_t$. However, technical difficulties arise with each term in this chain rule expansion. Below, we propose practical estimators for each term that overcome these difficulties.

**Gradient $\nabla_\phi \boldsymbol{r}_t$.** The difficulty here is differentiating through the expectation in Eq. (7), especially when $\boldsymbol{y}$ is a discrete random variable (integer counts in our later overdose or wildlife applications). We use the *score function trick* (Mohamed et al., 2020), popularized by Ranganath et al. (2014) yet dating back decades (Kleijnen & Rubinstein, 1996), sometimes also called REINFORCE (Williams, 1992). We can draw $M$ samples $\boldsymbol{y}_t^{(m)} \sim p_\phi$, then compute

$$\nabla_\phi \boldsymbol{r}_t = \frac{1}{M} \sum_{m=1}^{M} \frac{\boldsymbol{y}_t^{(m)}}{\mathbf{1} \cdot \boldsymbol{y}_t^{(m)}} \nabla_\phi \log p_\phi(\boldsymbol{y}_t). \qquad (10)$$

This estimator can reuse the $M$ i.i.d. samples already used to evaluate $\boldsymbol{r}_t$ in a forward pass. This is easy to implement for any model $p_\phi$ where sampling and evaluating the pdf is feasible, as we have assumed. We use automatic differentiation to compute $\nabla_\phi \log p_\phi(\boldsymbol{y}_t)$.

A downside of this estimator is high variance. We mitigate this with large $M$ values, though future work may use *control variates* (Ranganath et al., 2014; Mohamed et al., 2020) or try other estimators for gradients of discrete expectations (Maddison et al., 2017; Dimitriev & Zhou, 2021).

**Gradient $\nabla_{\boldsymbol{r}_t} L_t$.** For losses $L$ defined in terms of TOPKMASK binary vectors, like BPR, it is difficult to compute useful gradients because this loss is flat almost everywhere with respect to the input rankings $\boldsymbol{r}$. To overcome this barrier, we leverage recent advances in *perturbed optimization* (Berthet et al., 2020), also referred to as *stochastic smoothing* (Abernethy et al., 2016). A recent computer vision method (Cordonnier et al., 2021) shows how these ideas enable selecting a top-K set of patches from a high-resolution image for downstream prediction. We adapt this top-K approach to spatiotemporal forecasting for intervention.

Concretely, Cordonnier et al. (2021) obtain tractable $J$-sample Monte Carlo estimates of both the top-K indicator vector $\boldsymbol{b} = \text{TOPKMASK}(\boldsymbol{r}, K)$ and the Jacobian $\nabla_{\boldsymbol{r}} \boldsymbol{b}$ needed for backpropagation. First, we draw $J$ independent samples of a standard Gaussian noise vector of size $S$:

$\boldsymbol{z}_j \sim \mathcal{N}(0, I_S)$. Then, we compute

$$\hat{\boldsymbol{b}} = \frac{1}{J} \sum_{j=1}^{J} \boldsymbol{b}_j(\boldsymbol{r}), \quad \boldsymbol{b}_j(\boldsymbol{r}) = \text{TOPKMASK}(\boldsymbol{r} + \sigma \boldsymbol{z}_j).$$

$$\nabla_r \hat{\boldsymbol{b}} = \frac{1}{J\sigma} \sum_{j=1}^{J} \text{OUTER}(\boldsymbol{b}_j(\boldsymbol{r}), \boldsymbol{z}_j). \qquad (11)$$

The Jacobian $\nabla_{\boldsymbol{r}} \hat{\boldsymbol{b}}$ is an $S \times S$ matrix, where entry $j, k$ gives the scalar derivative $\frac{\partial b_j}{\partial r_k}$. The conceptual justification for the Jacobian estimator comes from Abernethy et al. (2016) and Berthet et al. (2020). Noise level $\sigma > 0$ is a hyperparameter that sets the strength of stochastic smoothing. It must be carefully selected in practice to add enough noise so that indicators $\boldsymbol{b}_j$ change for different samples $z_j$, but not too much noise so the $\boldsymbol{b}_j$ preserve the signal in $\boldsymbol{r}$.

Putting our score-function trick and perturbed optimizer estimators together, we compute the overall gradient of loss at index $t$ as a product of individual estimators: $\nabla_\phi L_t = \nabla_\phi \boldsymbol{r}_t \nabla_{\boldsymbol{r}_t} \boldsymbol{b}_t \nabla_{\boldsymbol{b}_t} L_t$. We compute the last term $\nabla_{\boldsymbol{b}_t} L_t$ via automatic differentiation.

Armed with this gradient estimator, we can pursue direct minimization of $\mathcal{J}^{\text{BPR}}$ via stochastic gradient descent methods. Stochasticity here comes from both $M$ score function samples and $J$ perturbation samples. For convenience and reliability, we use all $T$ records in the training set in every estimate, avoiding minibatching over time.

We assume the true data-generating distribution is unchanged across train and test time periods. If this does not hold, objectives that just average over $t$ as in Eq. (9) may have disadvantages. Instead we could upweight later $t$, or minimize out-of-sample error as in Gupta et al. (2024).

**Other methods for decision-aware training.** Our approach to direct BPR optimization here is an example of *decision-aware* or decision-focused training. In the taxonomy of Mandi et al. (2024), our approach is in the family of *differentiable perturbed optimizers*. Other work instead pursues *surrogate losses*. The *SPO+* method (Elmachtoub & Grigas, 2022) finds a convex surrogate for the "smart predict then optimize" optimization problem. The perturbed gradient (PG) method (Huang & Gupta, 2024) develops a more sophisticated surrogate, with theory and experiments suggesting utility even with misspecified models. In both cases, surrogate bounds make SGD-based learning tractable for a wide set of optimization tasks, including our knapsack-like BPR problem but also other tasks like shortest path finding.

**Downsides of only fitting BPR.** When models are misspecified, directly estimating $\phi$ to minimize $\mathcal{J}^{\text{BPR}}$ should yield better BPR than some $\phi'$ fit via conventional loss $\mathcal{J}^{\text{NLL}}$, and better BPR means better decisions. However, probabilistic forecasts of near-future *outcomes* $\boldsymbol{y}_{t+1}$ produced by BPR-trained $\phi$ have questionable utility. Nothing in the $\mathcal{J}^{\text{BPR}}$ objective makes $p_\phi$ accurately reconstruct even the train set $\boldsymbol{y}_{1:T}$; only *relative* ranking of sites matters. Even with

high-quality decisions, a model which produces unlikely forecasts may be difficult to interpret or verify.

### 4.3. Decision-aware maximum likelihood

To jointly achieve the goals of good top-K decisions and accurate forecasts across all sites, we propose to find parameters that solve a constrained optimization problem:

$$\underset{\phi}{\operatorname{argmin}} - \sum_{t=1}^{T} \log p_\phi(\boldsymbol{y}_t), \quad \text{s.t. } g_t(\phi) \leq 0 \,\forall\, t, \quad (12)$$

$$\text{where } g_t(\phi) = \epsilon + L^{\text{BPR}}(\boldsymbol{r}_t(\phi), \boldsymbol{y}_t).$$

Here, $\epsilon$ is a desired *lower bound* on a tolerable BPR for the decision task. Function $g_t(\phi)$ checks if the constraint is satisfied at time $t$, returning a non-positive value when $\text{BPR}_t \geq \epsilon$ and a positive value otherwise. A feasible solution $\phi$ must deliver BPR as good or better than $\epsilon$ on the provided training set. Practitioners can set $\epsilon$ to achieve a desired minimum value for BPR. For example, if BPR below 60% was unworkable to stakeholders, set $\epsilon = 0.6$ to enforce $0.6 \leq \text{BPR}$, recalling by definition $\text{BPR} = -L^{\text{BPR}}$.

We call this combined objective *decision-aware ML estimation*, or DAML. If the model is well-specified and training data are plentiful, DAML should deliver the same parameters as ML estimation when $\epsilon$ is low enough. However, when the model is misspecified and $\epsilon$ is higher than the BPR delivered by ML-estimated $\phi$, we argue DAML's constraint will produce better top-K decisions than ML alone, trading lower likelihood for higher BPR. Compared to direct minimization of $\mathcal{J}^{\text{BPR}}$, DAML can deliver similar BPR but more accurate forecasts of $\boldsymbol{y}$ for *all sites*. Additionally including the ML objective offers the ability to include an optional prior term $\log p(\phi)$ to incorporate any prior knowledge.

To solve in practice, we use the penalty method (Chong & Żak, 2013) to convert to an unconstrained loss:

$$\mathcal{J}^{\text{DAML}}(\phi) = \sum_{t=1}^{T} \lambda \max(g_t(\phi), 0) - \log p_\phi(\boldsymbol{y}_t). \quad (13)$$

This DAML formulation makes estimating $\phi$ via gradient descent possible. Here, $\lambda > 0$ is a nuisance hyperparameter that must be tuned. When the constraint is not satisfied, larger $\lambda$ values force gradient updates to move parameters further in directions that might satisfy the constraint. In practice, we set $\lambda$ such that both components of the loss are of similar magnitude during early training.

**Implementation.** Pseudocode for DAML training is provided in the supplement (Alg. A.1). There are several key hyperparameters. First, $M$ and $J$ are the number of Monte Carlo samples used during training to estimate gradients via the score function trick and perturbed optimizer method. Setting $M$ and $J$ larger produces lower variance estimates, but at the cost of runtime and memory. Consequently, we

recommend setting $M$ and $J$ as high as affordable. We found setting both to 100 worked for tasks in Sec. 5.

Proper selection of $\sigma$, the standard deviation of the Gaussian noise in the perturbed estimator, is also vital. If too small, estimated gradients will be zero; too large will swamp out any data-driven signal for learning. In practice, we found that $\sigma$ values a bit smaller than the largest elements of rating $\boldsymbol{r}^*(\phi)$ worked well. Because our ratio estimator produces values between 0 and 1, we set $\sigma$ between $10^{-3}$ and $10^{-1}$.

## 5. Experiments and Results

Decision-aware training can benefit a variety of models in diverse problem domains. The subsections below cover toy and real applications. On each task, we fit models using all three training approaches from Sec. 4: ML estimation, BPR optimization, and DAML. We wish to verify common hypotheses throughout: (i) ML training can yield suboptimal top-$K$ decisions; (ii) direct optimization of BPR improves this at the expense of likelihood; (iii) our DAML allow navigating tradeoffs between likelihood and BPR.

**Common setup.** In each task below, for a fixed task-specific $K$ we first train models for ML and BPR. Using their final BPR values as guidelines, we select a suitable range of $\epsilon$ values for DAML and fit each one, intending to explore intermediate points on the two-objective Pareto frontier (Costa & Lourenço, 2015). When training each objective, we run many random initializations to convergence across a range of learning rates and other hyperparameters (see App. C.2 for details). We keep the model $\phi$ from one run that best achieved its objective on a validation set, early stopping as needed. A model's ultimate top-$K$ rankings can have some stochasticity. Thus, later figures show *estimated distributions* of BPR across 1000 trials of a $M=1000$ sample Monte Carlo estimate of $\boldsymbol{r}(\phi)$ after training is completed.

### 5.1. Synthetic Data

We begin with an illustrative synthetic case study chosen to highlight the trade-offs that decision-aware training allows a modeler to make when working with misspecified models.

**Task.** We create a synthetic dataset of $S = 7$ sites over $T = 500$ times. Integer data $y_{ts}$ is generated i.i.d over time from a quantized Gaussian with site-specific mean and small constant variance. The first six sites have means evenly spaced between 10 and 60; the last site has a large mean of 100. Fig. 2 (top left) shows the training data.

**Model.** To show the benefits of our framework, we focus on a *misspecified model*. In particular, we model each site with a $L$-component positive Gaussian mixture model, with global mean and variance parameters and site-specific

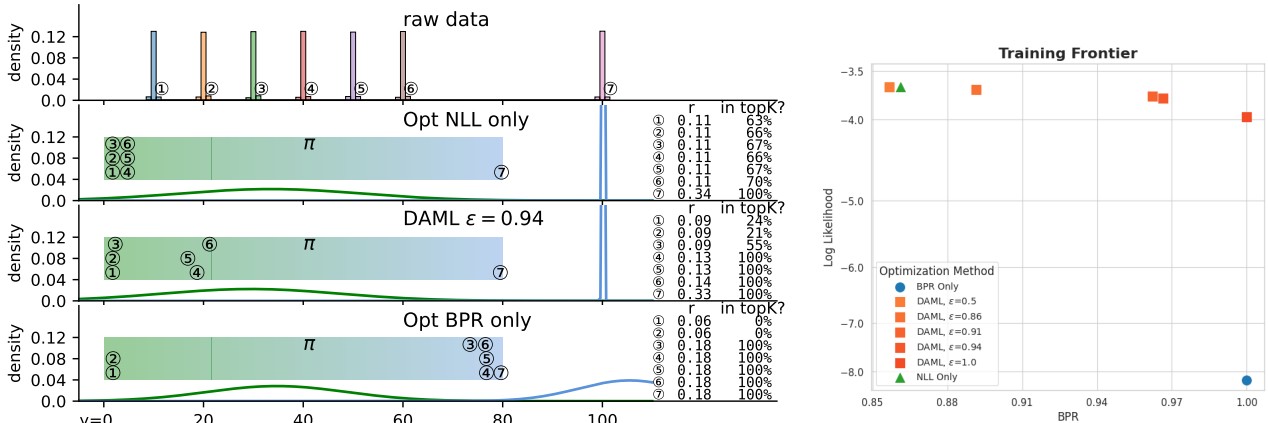

Figure 2: **Synthetic 1D data: learned models and Pareto frontier.** *Left Row 1:* histograms of $y_s$ values by site (circled numbers). Sites 3-7 should be the top K=5 under the true model. *Left Rows 2-4:* Learned Gaussian components, with site-specific weights $\boldsymbol{\pi}_s$ marked as horizontal position between pure green and blue. Text provides ranking $r$ with how often that site is in top $K = 5$ over 200 trials. *Right:* Likelihood vs. BPR tradeoff frontier for final models delivered by different training objectives.

component frequencies:

$$p_\phi(y_s) = \sum_{\ell=1}^{L} \pi_{s,\ell} \cdot \mathcal{N}_+(y_s | \mu_\ell, \sigma_\ell^2) \qquad (14)$$

Here $\phi = \{\mu_{1:L}, \sigma_{1:L}, \pi_{1:S,1:L}\}$. We reparameterize to unconstrained real values to make gradient-based learning possible: see App. D for details.

With $L=7$ components, the model would be well-specified and could recover the true data-generating process. However, we focus on misspecification, so we fit with $L=2$ components. This will hurt likelihood performance, as sites with distinct true means will need to use common means. However, we wish to show that solid top-K decision-making can still happen even with such severe misspecification.

**Experiment setup.** We use BPR with $K = 5$ for this task. For reproducible details, including all hyperparameters, see App. D. We only report training set metrics here for simplicity; later tasks assess generalization to test data.

**Results and analysis.** From results in Fig. 2, we draw several conclusions. First, training to optimize NLL alone delivers subpar BPR for this task. Second, optimizing for BPR alone yields much better BPR values, suggesting that even this mispecified model can deliver much better top-K decision making than the off-the-shelf ML solution. However, BPR alone yields nonsensical likelihood values, as nothing in the objective forces $\phi$ to be good at modeling the outcomes $y$, only at relative ranking of the $S$ sites. In Fig. 2, we see how our DAML hybrid objective allows a user to traverse the Pareto frontier of likelihood and BPR by enforcing a desired threshold on minimum BPR. As the desired minimum BPR threshold $\epsilon$ increases, we can sweep the tradeoff between likelihood and BPR. Ultimately, our DAML yields the best high-likelihood, high-BPR solutions in the top-right corner of the Pareto plot.

### 5.2. Opioid-related Overdose Forecasting

**Motivation.** The ongoing opioid overdose epidemic in the United States has incurred over 500,000 deaths in the past decade, with more than 80,000 fatal opioid-related overdoses in 2023 alone (Ahmad et al., 2025). Possible evidence-based interventions to mitigate overdose fatalities include overdose education and nalaxone distribution. Scarce resources require local decision makers to allocate these interventions to *small areas* that are *high-risk* (Allen et al., 2024), with co-incident education and support for proper follow-through. Public health agencies could use forecasting to help allocate limited resources towards the goal of harm reduction.

Several efforts have developed small-area forecasting models of opioid-related events (Marks et al., 2021b; Neill & Herlands, 2018; Bauer et al., 2023). A preregistered trial for overdose reduction in Rhode Island (Marshall et al., 2022) used BPR-like metrics to evaluate the top-$K$ predictions of conventionally-trained models. This past work does not *rank* or *train* to improve BPR, as we do.

**Datasets.** We study the capabilities of different methods on two datasets of historical opioid-related overdose mortality. Our IRB provided a Not Human Subjects Research determination for analysis of this decedent data. The first dataset, *MA Fatal Overdoses*, covers opioid-related overdose deaths in the state of Massachusetts from 2001-2021. This dataset is publicly available upon request from the MA Registry of Vital Records and Statistics. The second dataset, *Cook County IL Fatal Overdoses*, tracks opioid-involved overdose deaths in the greater Chicago area between 2015 and 2022. This is an open dataset obtained via the public website of the Cook County Medical Examiner Case Archive (Cook County, IL, 2014-present).

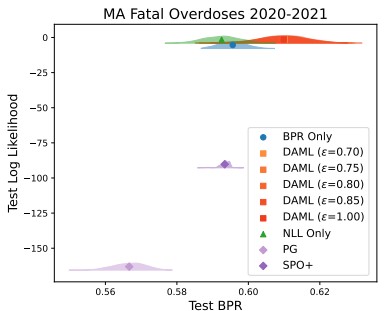 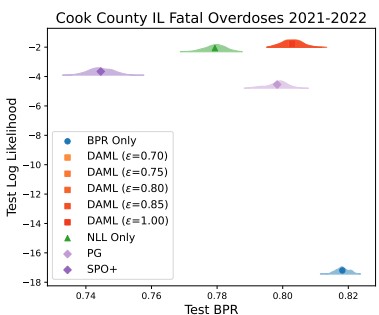 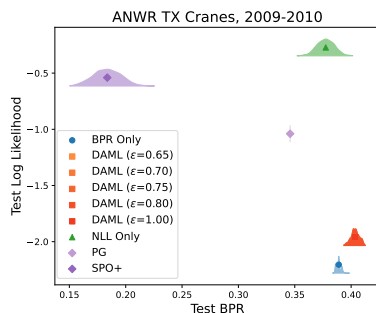

Figure 3: **Pareto frontier of best possible reach (BPR, x-axis) and log likelihood (LL, y-axis) for real-world tasks**. Higher is better on both axes. Each panel how the final models estimated by different training methods score on the test set of a forecasting task defined in Sec. 5. To capture the stochasticity of BPR due to our sampling-based ranking estimator, for each model we show an estimated density for BPR over 1000 trials. Uncertainty in this plot only corresponds to uncertainty in BPR, log likelihood is a point estimate. In all three tasks, our proposed decision-aware ML (DAML) delivers better top-K decisions as measured by BPR than ML estimation. DAML also delivers likelihood comparable to ML methods and *much better* than directly optimizing BPR. In the Cranes dataset, the DAML objective surprisingly offers better BPR than the BPR-only objective, although the magnitude of this difference is small and perhaps due to the small-scale and sparsity of this dataset.

We follow previous evaluations of these datasets in Heuton et al. (2024). Each dataset was processed to a common format of fatal overdose counts per time and spatial unit. For the spatial units, we chose *census tracts*. Each tract by design contains a mean population of 4000 people, a scale that allows capturing variation in overdoses at the neighborhood level. For temporal binning, we picked calendar years to reflect the frequency at which health agencies might enact policy changes. Summary facts are in Tab. C.1.

**Task.** Our forecasting task is to predict the next year's count of opioid-related fatal overdoses in each census tract. For *MA*'s $S$=1620 tracts, we train on data from 2011-2018, tune hyperparameters on validation data from 2019, and test on 2020 and 2021. For *Cook County IL*'s $S$=1328 tracts, we train on data from 2015-2019, tune on 2020, and test on 2021 and 2022. These splits follow Heuton et al. (2024).

Given a trained model, we assess heldout likelihood over all sites as well as BPR with $K$=100 to measure the model's where-to-intervene ranking. A $K$=100 budget was selected to reflect realistic public health budgets, and is similar to values used in other studies (Marshall et al., 2022).

**Model.** A recent benchmark (Heuton et al., 2024) compared many models designed for fatal overdose forecasting, including neural architectures with attention (Ertugrul et al., 2019) and more classical statistical models. A top-performing model is *negative binomial mixed-effects regression* (Marks et al., 2021a). The generative model can be expressed as

$$y_{st} \sim \text{NegBin}(\mu_{st}, q), \qquad (15)$$
$$\log(\mu_{st}) = \beta_0 + \boldsymbol{\beta}^T \mathbf{x}_{st} + b_{0s} + b_{1s}t.$$

Here, count $y_{st}$ is modeled as a Negative Binomial, where the log of the number of successes to stop at $\mu_{st}$ is a lin-

ear function of feature vector $\mathbf{x}_{st}$ as well as a site-specific intercept $b_{0s}$ and site-specific $b_{1s}$ weight on time $t$. The parameter $q$ is a probability of success shared by all sites.

Feature vector $\boldsymbol{x}_{st}$ includes tract $s$'s *overdose gravity*, a recent average of opioid-related overdose deaths in tracts spatially near to $s$, as well as measures of sociological vulnerability. See App. C.2.1 for details.

The overall parameters to estimate during training are $\phi = \{q, \beta_0, \beta, b_{0,1:S}, b_{1,1:S}\}$. To make constrained parameters amenable to gradient descent, we employ suitable one-to-one transforms to unconstrained spaces. Random effect weights $\boldsymbol{b}$ are regularized via a *prior* that assumes a zero-mean Normal distribution with learnable covariance parameters. Full details are provided in App. E.

**Competitor methods.** We compare to two other decision-aware methods discussed above: Perturbed Gradient (PG) (Huang & Gupta, 2024) and SPO+ (Elmachtoub & Grigas, 2022), as implemented in PyEPO software (Tang & Khalil, 2024). To be fair, each uses the same model $p_\phi$, the ratio estimator to rank sites, and the gradient of this estimator in Eq. (10). We conduct a hyperparameter search over learning rate (and perturbation noise for PG), selecting the model with the best loss value on validation data.

**Setup.** We followed the common setup described above. For reproducible details specific to overdose tasks, see App. C.2.

**Results.** Results on *test* data for both MA and Cook County IL tasks are shown in Pareto frontier plots in Fig. 3. For both datasets, we see that ML training yields suboptimal top-K decisions. Direct optimization of BPR can improve BPR, though gains on test data over ML vary (+0.04 on Cook County; less than +0.01 on MA). Direct BPR and surrogate

loss (SPO+, PG) solutions can yield *much worse* likelihood, as expected. The surrogate loss methods provide lower quality decisions than the DAML and direct BPR approach. Our DAML approach provides better top-K decisions than the ML approach, with no visible decay in likelihood.

## 5.3. Endangered Bird Forecasting

**Motivation**. In the 1940s, whooping cranes were almost completely extinct in the U.S., with only 20 existing in the wild (Cannon, 1996). Thanks to efforts over the years to preserve their population, there are roughly 650 wild cranes today. This key species is still listed as endangered in 2025.

A major flock of cranes, known as the Aransas-Wood population, winters at the Aransas National Wildlife Refuge (ANWR) along the Gulf Coast of Texas (Vartanian, 2023), while spending summers north in Canada. Ecologists wish to actively monitor this population. Regular aerial surveys of the Texas wintering region have been conducted for decades (Taylor et al., 2015). Using binned spatiotemporal data of sighting counts over time from these surveys, we wish to offer data-driven forecasts of where cranes may be found. This could help conservationists decide where to send future human monitors or where to place $K$ fixed-location cameras to efficiently track population health.

**Data**. We use raw data from Taylor et al. (2015), which digitized decades of aerial surveys of ANWR that marked individual crane sightings on paper maps. We processed this *ANWR TX Cranes* data into a common format of sighting counts over time and space, selecting bin sizes to support our goal of using the top-$K$ sites to improve monitoring. Quality cameras or binoculars that could be used to track cranes can reasonably capture a 1.5 meter tall whooping crane in a 250-meter radius. Therefore, for spatial units, we divided the ANWR into 1338 boxes, each 500 meters per side. For temporal binning, we use 2 month periods. This choice catches seasonal variation, but avoids how finer scales might burden staff to move cameras too often.

**Task**. The experiment on these data was trained on years 2002-2006, validated on 2007-2008, and tested on 2009-2010. In this context, a "year" represents one wintering season; winter 2010 means the winter that began in October 2010 and ended in April 2011. We choose $K = 50$ for this task. This is an estimate of how many sites scientists might reasonably monitor every two months on a limited budget.

**Models**. The same negative binomial mixed-effect model was used as in Sec. 5.2. Features $\boldsymbol{x}_{st}$ for site $s$ at time $t$ include historical bird counts at $s$, the latitude and longitude of the site's centroid, time of year, and the overall time.

**Setup**. We followed the common setup described above. See reproducible details in App. C.3.

**Results.** From results in Fig. 3 (right panel), we find that optimizing NLL or BPR *alone* will yield poor results in the other metric. In this case, direct optimization of BPR results in the best top-K decisions, but dramatically worse likelihood. Our DAML models improve BPR by up to 0.05 over conventional ML training with only modest decay in likelihood. Our DAML and direct BPR objectives also deliver better BPR than previous decision-aware methods (PG, SPO+). This is even after providing smarter initializations to these methods, as we found their training had trouble improving on our common random initialization of $\phi$, perhaps due to this task's much sparser $\boldsymbol{y}$ values.

## 6. Discussion

We have addressed two open challenges related to top-K resource allocation problems guided by the best possible reach (BPR) performance metric. We provided a ranking strategy that can outperform simple per-site means. We posed a training objective that strives for high likelihood across all sites while ensuring top-K decisions meet a stakeholder-specified quality level. Our experimental evaluations suggest our approach can better manage tradeoffs in likelihood and BPR than conventional training methods or previous decision-aware methods.

There are several limitations to this study. We focused on showing the tradeoffs between likelihood and BPR for a fixed model family in each task without comparing a wide variety of possible models. Only one type of model misspecification is considered each in the synthetic and real-world experiments; perhaps decision-aware objectives are more or less different to maximum likelihood estimates depending on the kind of model misspecification. Our evaluations use fixed $K$ values and do not explore sensitivity to $K$ or other hyperparameters. Practitioners may need multiple metrics to assess overall utility; focusing myopically on BPR may not always be wise. Additionally, our framework assumes any site with large outcome $y_s$ is a better candidate for intervention. Future work could explore data-driven site selection that considers how site-specific attributes might make some interventions more or less effective.

Looking forward, we hope to see applications of these ideas inform public health, wildlife conservation, and other where-to-intervene decision-making problems across the private and public sectors. We also hope that future methodological work could scale up to much larger problems ($S > 2000, T > 20$) as we found that our current experiments tested the limits of commodity GPUs.

## Acknowledgments

Authors KH, FSM, and MCH are supported in part by the U.S. National Science Foundation (NSF) via grant IIS #

2338962. Author TJS was supported by the U.S. National Institute on Drug Abuse via grant # R01DA054267. Our team also acknowledges support from the American Public Health Association for a data science demonstration project. We are thankful for computing infrastructure support provided by Tufts University, with hardware funded in part by NSF award OAC CC* # 2018149.

We are grateful to several anonymous reviewers, who pointed us to related work in operations research literature and helped us connect our BPR metric to work on knapsack problems.

## Impact Statement

Recent work on where-to-intervene decision making has emphasized the importance of incorporating additional constraints into the top-$K$ site selection budget to meet the needs of all constituents. For example, Marshall et al. (2022) sought a balance of rural and urban sites for intervention in overdose mitigation efforts in the state of Rhode Island. We believe exciting methodological work could extend the ideas in this paper to handle such constraints.

We also acknowledge that our decision-aware methods are currently technically more complex than existing off-the-shelf solutions, presenting barriers to adoption in many downstream applications where practitioners have limited hardware and limited expertise. We hope to work with stakeholders to build open-source packages that would allow DAML models to be created with off-the-shelf ease on a new dataset of interest.

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

Appendix

This appendix includes additional experimental results and information for understanding and reproducing experiments.

Reproducible code used for all experiments is included in an open-source repository:

https://github.com/tufts-ml/decision-aware-topk/

# Contents

## A. Pseudocode for Training

---

**Algorithm A.1** Decision-aware ML training

---

**Input**:

• $\{\mathbf{y_t}\}_{\mathbf{t=1}}^{\mathbf{T}}$, train data, each $\boldsymbol{y}_t \in \mathbb{R}_{\geq 0}^S$

• $\phi \in \mathbb{R}^P$: parameter vector for model

• $M$: int num MC samples for score func estimator

• $J$: int num MC samples for stochastic smoothing estimator

• $\sigma > 0$ : float stddev of stochastic smoothing estimator

• $\epsilon \in (0,1)$ : Desired minimum BPR value. Will try to enforce constraint BPR $\geq \epsilon$.

• $\lambda > 0$ : Strength multiplier when constraint is violated.

**Output**: Trained model parameter $\phi$

**Procedure**:

1: **while** not converged **do**
2:     $\nabla_\phi \mathcal{J} \leftarrow \mathbf{0}$                 // $P \times 1$ vector to store grad wrt params
3:     **for** time $t \in \{1, 2, \ldots T\}$ **do**
4:        $\{\mathbf{y}_t^m\}_{m=1}^M \sim p_\phi$                // $M$ Monte Carlo (MC) samples
5:        $\mathbf{r}_t \leftarrow \frac{1}{M} \sum_m \frac{\boldsymbol{y}_t^m}{\mathbf{1} \cdot \boldsymbol{y}_t^m}$             // $S \times 1$ ranking vector via MC
6:
7:        $\mathbf{b}_t \leftarrow \text{TOPKMASK}(\mathbf{r}_t)$
8:        $L_t^{\text{BPR}} \leftarrow -\frac{1}{\boldsymbol{y}_t \cdot \text{TOPKMASK}(\boldsymbol{y}_t)} (\boldsymbol{y}_t \cdot \mathbf{b}_t)$    // Scalar loss, -BPR
9:        $g_t^{\text{BPR}} \leftarrow \epsilon + L_t^{\text{BPR}}$            // Scalar. Negative if BPR $\geq \epsilon$ is satisfied.
10:       $\mathcal{J}_t^{\text{BPR}} \leftarrow \lambda \max(g_t^{\text{BPR}}, 0)$      // Scalar ultimate loss for BPR
11:
12:       $\nabla_\phi r_t \leftarrow \frac{1}{M} \sum_m [\nabla_\phi \log p_\phi(\boldsymbol{y}_t^m)] \frac{\boldsymbol{y}_t^m}{\mathbf{1} \cdot \boldsymbol{y}_t^m}$    // $P \times S$ matrix, score func. est. of $\nabla_\phi r_t$
13:       $\{\mathbf{z}_j\}_{j=1}^J \sim \mathcal{N}(0, I_S)$
14:       $\nabla_{r_t} b_t \leftarrow \frac{1}{\sigma} \frac{1}{J} \sum_{j=1}^J \text{OUTER}(\text{TOPKMASK}(\mathbf{r}_t + \sigma \mathbf{z}_j), \mathbf{z}_j)$    // $S \times S$ matrix, perturbed estimate of $\nabla_{r_t} b_t$
15:       $\nabla_{b_t} \mathcal{J}_t^{\text{BPR}} \leftarrow -\lambda \frac{1}{\boldsymbol{y}_t \cdot \text{TOPKMASK}(\boldsymbol{y}_t)} \boldsymbol{y}_t \cdot \mathbf{1}[g_t^{\text{BPR}} > 0]$    // $S \times 1$ vector, nonzero if constraint unsatisfied.
16:       $\nabla_\phi \mathcal{J}_t^{\text{BPR}} \leftarrow (\nabla_\phi r_t)(\nabla_{r_t} b_t)(\nabla_{b_t} \mathcal{J}_t^{\text{BPR}})$    // $P \times 1$ vector
17:
18:       $\mathcal{J}_t^{\text{NLL}} \leftarrow -\log p_\phi(\boldsymbol{y}_t)$        // Scalar ultimate loss for NLL
19:       $\nabla_\phi \mathcal{J}_t^{\text{NLL}} \leftarrow -\nabla_\phi \log p_\phi(\boldsymbol{y}_t)$
20:
21:       $\nabla_\phi \mathcal{J} \leftarrow \nabla_\phi \mathcal{J} + \nabla_\phi \mathcal{J}_t^{\text{NLL}} + \nabla_\phi \mathcal{J}_t^{\text{BPR}}$
22:     **end for**
23:     $\phi \leftarrow \text{GRADDESCENTUPDATE}(\phi, \nabla_\phi \mathcal{J})$      // Use steepest descent or Adam or ...
24: **end while**
25: **return** $\phi$

---

## B. Ranking Demo for BPR

### B.1. Justification for ratio estimator

In the text, we propose the ratio estimator for ranking locations: $\boldsymbol{r} = \mathbb{E}[\frac{\boldsymbol{y}}{\mathbf{1} \cdot \boldsymbol{y}}]$ and justify its use by demonstrating that it is a better bound on our decision loss BPR than the mean estimator. However, a natural question is, why not use a ranking estimator which more closely resembles BPR, such as $\boldsymbol{r} = \mathbb{E}[\frac{\boldsymbol{y}}{\text{TopKMask}(\boldsymbol{y}, K)} \cdot \boldsymbol{y}]$. In practice, this estimator is equivalent to the ratio estimator in expectation, and will select the same top-K locations. Accordingly, we use the ratio estimator for its improved computational speed and simplicity.

## B.2. Demo experiment

To gain understanding about the problem of *ranking* to optimize the fraction of best possible reach (BPR) performance metric, here we present detailed analysis of a toy problem with $S = 9$ sites. We'll assume the true data-generating process for each site is completely known throughout and that each site can be modeled independently of other sites.

$$p(\boldsymbol{y}_{1:9}) = \prod_{s=1}^{9} p(y_s) \tag{16}$$

For each site, we select from 3 possible site-specific model archetypes, nicknamed type A, type B, and type C.

- sites #1, #2, and #3 are each iid with type A PMF

- sites #4, #5, and #6 are each iid with type B PMF

- sites #7, #8, and #9 are each iid with type C PMF

Each type's PMF function over the non-negative integers is defined in the table below.

|  | Type A | Type B | Type C |
|---|---|---|---|
|  | $p(y_s) = \begin{cases} 0.0 & \text{if } y_s = 0 \\ 1.0 & \text{if } y_s = 7 \\ 0.0 & \text{otherwise} \end{cases}$ | $p(y_s) = \begin{cases} 0.35 & \text{if } y_s = 0 \\ 0.65 & \text{if } y_s = 10 \\ 0.0 & \text{otherwise} \end{cases}$ | $p(y_s) = \begin{cases} 0.90 & \text{if } y_s = 0 \\ 0.10 & \text{if } y_s = 80 \\ 0.0 & \text{otherwise} \end{cases}$ |
| Mean | 7.0 | 6.5 | 8.0 |
| Median | 7.0 | 10.0 | 0.0 |

We have now defined the joint PMF $p(\boldsymbol{y}_{1:9})$ over the 9 sites.

We can compare two possible ways to compute a numerical ranking of the $S = 9$ sites:

- **Mean estimator**, which computes the per-site mean: $\boldsymbol{r} = \mathbb{E}[\boldsymbol{y}]$

- **Ratio estimator**, which computes the expectation of $\boldsymbol{y}$ normalized by its sum: $\boldsymbol{r} = \mathbb{E}[\frac{\boldsymbol{y}}{\mathbf{1} \cdot \boldsymbol{y}}]$

For each possible ranking strategy, we repeated BPR calculations over 10000 trials. To ensure accuracy of Monte Carlo estimates, we average over $M = 50000$ samples to estimate the expectation defining each $\boldsymbol{r}$.

Results are provided in the table below. We have two key findings. First, our proposed ratio estimator can select very different sites than the per-site mean, *even when both estimators have access to the true data-generating model.* Using $K = 3$, our estimator would select all type-A sites as the top 3; in contrast the per-site mean would select all type-C sites (#7-9). Second, this can produce very different BPR values. Even in this simple example, we see an absolute difference in BPR of over 0.4 between the different rankings at $K = 1$ and over 0.39 at $K = 3$, which is a huge shift for a metric that is bounded between 0.0 and 1.0.

|  |  | BPR | | | fraction of trials each site in top $K$=3 of $\boldsymbol{r}$ | | | | | | | | |
|---|---|---|---|---|---|---|---|---|---|---|---|---|---|
|  |  | $K$=1 | $K$=3 | $K$=6 | A:#1 | A:#2 | A:#3 | B:#4 | B:#5 | B:#6 | C:#7 | C:#8 | C:#9 |
| mean | $\boldsymbol{r} = \mathbb{E}[\boldsymbol{y}]$ | 0.107 | 0.231 | 0.636 | 0.0 | 0.0 | 0.0 | 0.0 | 0.0 | 0.0 | 1.0 | 1.0 | 1.0 |
| ratio | $\boldsymbol{r} = \mathbb{E}[\frac{\boldsymbol{y}}{\mathbf{1} \cdot \boldsymbol{y}}]$ | 0.538 | 0.625 | 0.810 | 1.0 | 1.0 | 1.0 | 0.0 | 0.0 | 0.0 | 0.0 | 0.0 | 0.0 |

These results can be replicated via scripts provided in the code repository.

| | # Spatial Sites | Temporal Scale | Outcomes $y$ | Features |
|---|---|---|---|---|
| MA Fatal Overdoses | 1620 Census Tracts | 20 years, 2001-2021 | Count of opioid-related fatal overdoses | SVI, Past Deaths, Location, Time |
| Cook County IL Fatal Overdoses | 1328 Census Tracts | 8 years, 2015-2022 | Count of opioid-related fatal overdoses | SVI, Past Deaths, Location, Time |
| Aransas TX Whooping Cranes | 1338 boxes, each 500m×500m | 60 years, 1952-2011 (last 10 years used) | Count of bird spotted in aerial survey | Past observations, Location, Time, Month |

Table C.1: Comparison of Real datasets, in terms of number of spatial sites $S$, temporal scales, outcomes $y$, and features.

| ANWR TX Cranes | MAE | RMSE | BPR-50 |
|---|---|---|---|
| Last timestep | 0.24 | 1.04 | 0.27 |
| Avg over 10 | 0.25 | 0.78 | 0.38 |
| Chance decision, $\hat{y} = 0$ | 0.15 | 0.79 | 0.05 |
| EpiGNN | 0.38 | 0.70 | 0.06 |
| PG | 1.03 | 4.85 | 0.35 |
| SPO+ | 0.15 | 0.83 | 0.18 |
| NLL Only | 0.35 | 1.14 | 0.38 |
| DAML ($\epsilon = 1$) | 9.08 | 35.89 | 0.39 |
| BPR Only | 40495 | 40495 | 0.41 |

(a) ANWR TX Cranes 2009-2010.

| Cook County IL Overdose | MAE | RMSE | BPR-100 |
|---|---|---|---|
| Last timestep | 1.07 | 1.66 | 0.76 |
| Avg over 5 | 0.99 | 1.58 | 0.80 |
| Chance decision, $\hat{y} = 0$ | 1.37 | 2.46 | 0.20 |
| EpiGNN | 1.33 | 2.03 | 0.32 |
| PG | 5.65 | 17.03 | 0.80 |
| SPO+ | 1.25 | 2.38 | 0.74 |
| NLL Only | 1.03 | 1.58 | 0.78 |
| DAML ($\epsilon = 1$) | 1.12 | 1.84 | 0.80 |
| BPR Only | 70.31 | 152.72 | 0.82 |

(b) Cook County IL 2021-2022.

Table C.2: Performance comparison between different model families for error-based and decision metrics. The top 2 rows represent simple historical baselines: using either the previous timestep alone, or an average over a larger amount (10 bi-months for the crane dataset, and 5 years for Cook County). Next, the *Chance decision* model chooses spatial locations by random chance, and uses all 0's for predicted $\hat{y}$ in MAE and RMSE calculations. Despite the simplicity, this is a competitive model on the error-based metrics due to sparsity. It outperforms all others on MAE on the Cranes dataset. Next is EpiGNN, a spatiotemporal forecasting model based on graph neural networks, trained to minimize RMSE. Although this model has the best performing RMSE on the crane dataset, it makes poor top-K decisions on both. Finally, the last three rows show different ways to train the negative binomial mixed effects regression model in 14. *NLL Only* trains to maximize likelihood alone (8), *BPR Only* is our direct-loss minimization trained to maximize BPR alone (9), and DAML (13) is our hybrid loss that allows a user to explore the pareto frontier between likelihood and BPR-K.

# C. Experimental Details and Results from Real-World Data

## C.1. Results on alternative models and metrics

## C.2. Opioid-related Overdose Forecasting Results

### C.2.1. FEATURES

The feature vector $\boldsymbol{x}_{st}$ for site $s$ and time $t$ includes:

- Latitude and Longitude of the centroid of the census tract $s$

- The current timestep $t$

- Overdose gravity, an weighted average of the prior year's overdose deaths in all contiguous tracts. We construct the feature in the same way as (Marks et al., 2021a). Note that the original paper describes a weighted average over all regions within a radius, but the provided code uses only immediately contiguous locations. We follow the implementation from the code.

- Covariates from the Social Vulnerability Index (SVI) (CDC ATSDR, 2022). These covariates are available as 5-year estimates for every census tract in the United States and are updated every 2 years. The SVI measures report the

percentile ranking of every census tract according to 4 themes: Socioeconomic, Household Composition & Disability, Minority Status & Language, and Housing Type & Transportation. We use 5 variables: each tract's ranking in each of the four themes as well as its composite ranking.

- We include 5 temporal lags: the number of fatal opioid-related overdoses in tract $s$ in each of the past 5 years.

### C.2.2. HYPERPARAMETER RANGES

Hyperparameter ranges explored include:

- Perturbation noise: $0.1, 0.01$, and $0.001$.

- Adam step size: $0.1, 0.01$, and $0.001$.

- Number of samples for score function trick estimator: 100

- Number of samples for perturbation estimator: 100

- BPR constraint $\epsilon$: 5 possible values of the penalty threshold: $1.0$, for a threshold that always encourages better BPR, as well as 4 values selected to be around the best BPR obtained on the training data.

- Multiplier on the penalty for DAML: 30. This value was chosen so that the BPR and likelihood components were roughly the same magnitude after 100 epochs of training.

For each location, training objective (likelihood, direct loss minimization, and DAML), as well as for each threshold for the hybrid model, we selected the model based on validation dataset performance. For the maximum likelihood model and direct BPR models we picked the model that best maximized their respective objective. For the DAML models, we found that given the larger number of hyperparameter configurations, small amount of validation data, and BPR's sensitivty, selected a model based on BPR lead to overfitting. To ameliorate this, we chose the model with the highest likelihood provided that the BPR was greater than a given threshold. Because models failed to meet the target $\epsilon$, we chose the BPR of the maximum likelihood model on the validation dataset as our threshold. If no models met this threshold, we selected the maximum BPR model. For the We did this by evaluating the validation performance every 10 epochs, and saving a checkpoint for the model with the lowest loss on validation data.

### C.3. Endangered Bird Forecasting Results

#### C.3.1. FEATURES

The feature vector $\boldsymbol{x}_{st}$ for site $s$ and time $t$ includes the following variables: the past 5 count values at site $s$, latitude & longitude of site $s$'s centroid, an enumerated timestep indicating time passed at $t$ since the starting timestamp of the dataset, and a monthly indicator variable. The dataset includes three bimonthly periods per wintering season: `Oct 20-Dec 24`, `Dec 25-Feb 27`, and `Feb 28-Apr 30`. The monthly indicator received a value of 1, 2, or 3, respectively, depending on which set of months an observation spanned.

#### C.3.2. HYPERPARAMETER RANGES

The same hyperparameter and model selection was performed on Whooping Crane data as the opioid experiment in Section C.2.

## D. Experimental Details and Results from Synthetic Data

**Model details.** As explained in the main paper, we use a mixture of $L = 2$ Gaussian distributions where each is *truncated* to the positive reals. We give each of the $S$ locations their own mixture weights $\pi_{s,l}$ where $\sum_{l=1}^{2} \pi_{s,l} = 1$ and $\pi_{s,l} \geq 0$. Our model for an individual location is then:

$$p(y_s) = \sum_{l=1}^{2} \pi_{s,l} \cdot \mathcal{N}_+(y_s | \mu_l, \sigma_l^2) \tag{17}$$

Our parameter vector $\phi$ then consists of the set of all $\mu_l, \sigma_l$ and $\pi_{s,l}$. We have that both $\mu_l$ and $\sigma_l$ should be positive, as we are modeling positive counts and standard deviation is defined to positive. To accomplish this, we transform these variables using the softplus function when performing gradient based learning. Additionally, we constrain $\sigma_l \geq 0.2$ to avoid degeneracy. Finally, to make a valid pdf, we have that the mixture weights must sum to one: $\sum_{l=1}^{L} \pi_{s,l} = 1$. To enforce this, we transform unconstrained variables using the softmax transform. This creates an subtle issue with model identifiability, as there are many unconstrained values that will lead to similar weights, but we do not find that this impacts our ability to train models effectively to achieve good likelihood or good BPR with the appropriate objective.

**Training Procedures.** We seek to train 7 models: one that optimizes only for model log likelihood, one that optimizes only BPR-5, and 5 models penalized to achieve certain threshold values of BPR. Each model is given 20 random initializations of the model parameters. We use a learning step-size of 0.1. For models with BPR in the objective, we try a perturbation noise of both 0.01 and 0.05, with 500 samples. In the hybrid models, we use $\lambda = 30$, selected so that the likelihood term and the BPR penalty term are on the same order of magnitude after several epochs of training.

**Tradeoffs between likelihood and BPR when L=2.** If this model had $L = 7$ components, it would be well-specified and recover the true data generating process. However, with only 2 components, it is forced to group locations with distant mean values, which will come at a cost to likelihood and predictive capability as assessed by BPR.

For this example, we will consider BPR-5 as our decision making metric. A model that correctly ranks the top-5 locations will achieve perfect BPR. Our misspecified model is capable of this by learning 2 distinct mean values $\mu_k$, one higher than the other. As long as the mixture weights for the top-5 locations $\pi_s$ assign all probability to the high component, and the mixture weights for the bottom 2 locations assign all probability to the low component, the model will have perfect BPR.

However, this is not what the model with the best possible likelihood looks like. To maximize likelihood, a model will assign the 6 low locations to one component, and the one high location to another, as in Fig. 2.

Our hybrid objective DAML can explore the Pareto frontier between maximizing for likelihood and BPR-5. By including more locations into the high-valued component, BPR-5 will increase as log likelihood slightly decreases. Our hybrid objective formulation allows us to control this tradeoff by specifying the threshold at which the penalty term takes effect.

**Results.** Results are shown in Figure 2 in the main paper. Here we see the ability of the hybrid Decision-aware object to traverse the Pareto frontier between the best possible likelihood and BPR. The lowest threshold of 0.5 is trivially satisfied by the maximum likelihood model. The highest threshold of 1.0 is only satisfied by perfect BPR, while the 3 intermediate thresholds were chosen to explore the solution frontier that is possible by including or excluding a particular component from one of the learned mixtures. We see that by using the decision-aware training objective, we can maximize BPR while still obtaining a highly likely model.

## E. Model for Negative Binomial Mixed-Effects

Here we provide further detail on the model described in Eq. (15).

For some elements of the parameters $\phi$, we have a prior that informs point estimation in MAP fashion. The random effects are assumed to follow a multivariate normal prior

$$\begin{pmatrix} b_{0s} \\ b_{1s} \end{pmatrix} \sim \mathcal{N}\left( \begin{pmatrix} 0 \\ 0 \end{pmatrix}, \boldsymbol{\Sigma} \right) \tag{18}$$

where the covariance matrix $\boldsymbol{\Sigma}$ is parameterized as:

$$\boldsymbol{\Sigma} = \begin{pmatrix} \sigma_0^2 & \rho\sigma_0\sigma_1 \\ \rho\sigma_0\sigma_1 & \sigma_1^2 \end{pmatrix} \tag{19}$$

Here, $\sigma_0$ is the standard deviation of the random intercepts, $\sigma_1$ is the standard deviation of the random slopes, and $\rho$ is the correlation between the random intercepts and slopes, where $-1 \leq \rho \leq 1$. We pack these hyperparameters into a separate vector $\eta = \{\sigma_0, \sigma_1, \rho\}$.

**Transforming to unconstrained parameters.** To ensure parameter constraints are satisfied throughout gradient descent optimization, we employ invertible transformations that map constrained domains to unconstrained spaces. We reparameterize the correlation coefficient $\rho \in (-1, 1)$ using $u \leftarrow \text{arctanh}(\rho)$, allowing $u$ to be optimized over $\mathbb{R}$ while ensuring

$\rho = \tanh(u)$ remains within its valid bounds. For the strictly positive parameters $\sigma_0, \sigma_1 > 0$, we optimize unconstrained parameters $\xi_0, \xi_1 \in \mathbb{R}$ and apply the softplus transformation $\sigma_i \leftarrow \ln(1 + e^{\xi_i})$, which guarantees positivity while maintaining smoothness. Similarly, for the probability parameter $q \in (0, 1)$, we optimize an unconstrained parameter $\zeta \in \mathbb{R}$ and apply the sigmoid transformation $q \leftarrow 1/(1 + e^{-\zeta})$. During gradient descent, we optimize these unconstrained parameters, and transformed to the constrained version during forward sampling or pdf evaluation of the model.

**MAP estimation.** Together, the parameters $\phi$ and hyperparameters $\eta$ comprise this model. Both are point estimated to maximize the MAP objective. Thus, what is marked in the paper as NLL optimization is really best viewed as MAP or penalized NLL estimation, where the loss is

$$\mathcal{J}(\phi, \eta) = -\log p(\boldsymbol{y}_{1:T}|\boldsymbol{\phi}) - \log p(\phi|\eta) \tag{20}$$

Similarly, when we fit this model with DAML, the above loss is what is minimized subject to the BPR constraint.

