# OpenReview forum: "Decision-aware Training of Spatiotemporal Forecasting Models to Select a Top-K Subset of Sites for Intervention"
_ICML.cc/2025/Conference — ICML 2025 poster_

### Official Review · Reviewer_His4 · 2025-03-11

**Overall Recommendation:** 4

**Summary:**

In a spatiotemporal forecasting setting, the authors consider traning prodiction models adapted to the task of selecting the top-K sites for intervention. The authors consider BPR as the desired metric and develop an algorithm for training models using a gradient-based approach. The main difficulty of this approach lies in the uninformative gradients that the BPR objective, combined with the top-K selection provides. To differentiate through the sampling of discrete variables $r$, the authors adapt a REINFORCE-like approach, while deriving the Loss (which includes the top-K ranking) is done using a stochastic smoothing approach. In a last step, a combination of DPR and log-likelihood is used to propose DAML, which is intended to train decision aware models which provide good likelihood estimates.

**Claims And Evidence:**

The claims are well supported.

**Essential References Not Discussed:**

I think it would be valuable to discuss the connection to other works in the literature related to decision-aware learning, end-to-end learning and contextual optimization, see for instance Sadana et al. (2024) and references therein.

Sadana, Utsav, et al. "A survey of contextual optimization methods for decision-making under uncertainty." *European Journal of Operational Research* (2024).

**Experimental Designs Or Analyses:**

As far as I can tell, the experimental designs make sense.

**Methods And Evaluation Criteria:**

The evaluation methods are suitable.

**Other Comments Or Suggestions:**

The uncertainty plots in Figure 3 are quite confusing. Do they provide uncertainty only in the BPR direction? If not, why does uncertainty increase for lower log-likelihood values?

**Other Strengths And Weaknesses:**

The authors tackle an interesting problem and propose a new training scheme which is well adapted to the the problem and improves performance compared to the baselines. The paper is well written and explained, I specifically appreciated the interesting, and practically relevant experiments section.

**Questions For Authors:**

1. On line 91ff, right side, you state that the framework is flexible and can be applied to neural networks. How would you propose to sample from the joint model $p_\phi$ if it was parametrized by a multilayer neural network?
2. In Figure 2, left panel, “OPT BPR only”. I think this result is quite intersting, especially looking at the weights $\pi$. The model identifies the IDs to select (high $\pi$) and the ones not to select (low $\pi$). Intuitively, one would expect this to carry over to any top-K setting: No matter the complexity of the task, a mixture of 2 gaussians should be enough as it is a proxy for identifying the selected IDs (being identified with the gaussian with higher mean) and non-selected IDs (being identified with the gaussian with higher mean). Is this intuition true?
3. In Figure 2, on the right panel: DAML attains the same BPR as directly training on the BPR loss, but with better log-likelyhood. The structure of the BPR loss in equation (4) suggests that minimizing (9) could have multiple global optima. I think one view DAML (with appropriately chosen $\varepsilon$ as “finding the global optima of $\min(9)$ that has the highest log likelihood.
4. On line 252ff, left side: “For convenience and reliability, we use all T records in the training set in every estimate, avoiding minibatching over time.” Do the results remain similar if minibatching is used? I understand that for the datasets used minibatching may not be neccessary, but for larger datasets one would certainly like to use it.

**Relation To Broader Scientific Literature:**

The authors use a decision aware objective for training a model adapted to the top-K selection problem. I am not aware of any other works that tackle decision aware training for the top-K problem. Decision aware training is however a growing trend in the literature, see Sadana et al. (2024).

**Theoretical Claims:**

I followed the mathematical derivation in the main text, which looks correct to me.

---

> ### Author Rebuttal · Authors · 2025-04-01
>
> We thank the reviewer for their constructive comments about our work. We try to address key points below.
>
> > RE Question about “the title of Algorithm A.1”
>
> Algorithm A.1 is meant to summarize the decision-aware ML training (DAML) approach described in Sec. 4.3. Current title is “Decision-aware ML training for top-K tasks judged by BPR”. We will revise to just say “Decision-aware ML training” in case the latter half of the title was confusing
>
> ## Essential References (Common Issue)
>
> > I think it would be valuable to discuss the connection to other works in the literature related to decision-aware learning, end-to-end learning and contextual optimization
>
> We agree there’s a broader set of related work, especially from the OR community. We plan to cite and discuss the survey by Sadana et al. (2024), as well as valuable works pointed out by ZTv9.
>
> Specifically, we plan revisions that will
> * cite and discuss the application of “decision-aware denoising” to the problem of where to place speed humps to calm traffic (Gupta et al., SSRN preprint from 2024)
> * cite and discuss the health supply chain task of Chung et al. (ML4H 2022)
> * cite and discuss the PyEPO package (Tang & Khalil '24), especially how it can support solving problems like ours
> * cite the survey on “decision-focused learning” by Mandi et al. (2024) and the survey by Sadana et al. (2025), and relevant references therein
>
> > uncertainty plots in Figure 3 are quite confusing. Do they provide uncertainty only in the BPR direction?
>
> Yes, the uncertainty in this plot is only in the BPR direction. We will revise the caption to clarify.
>
> In Figure 3, we intend to show how each method’s estimated parameter performs across two metrics: log likelihood and BPR. Log likelihood is well-summarized by one deterministic number (vertical position of the marker). In contrast, BPR is not deterministic, as it depends on calculating the ratio estimator for r in Eq 7 which requires an average over M *samples* of y. Each method in Fig 3 has BPR visualized as a histogram showing the distribution across 1000 trials, with each trial using M =1000 samples.
>
> > 1. … How would you propose to sample from the joint model p if it was parametrized by a multilayer neural network?
>
> There’s a substantial literature on deep probabilistic models, which combine neural networks for flexible parameterization with classic statistical distributions with well-known sampling routines.
> For example, suppose our joint model for vector y was multivariate Gaussian with some mean and some covariance. The mean vector could be parameterized as the output of a neural network. So could the covariance matrix, assuming symmetry and positive definite constraints were enforced. Sampling from a Gaussian given its mean and covariance is then straightforward.
>
> > 2. In Figure 2, left panel, “OPT BPR only”. I think this result is quite interesting, … one would expect this to carry over to any top-K setting: No matter the complexity of the task, a mixture of 2 gaussians should be enough … for identifying the selected IDs (being identified with the gaussian with higher mean) and non-selected IDs … Is this intuition true?
>
> Yes we agree with this intuition. Deliberate construction of component weights $\pi$ for each site to assign the top K sites to a “high mean” component and other sites to a “low mean” component should be enough to get good BPR on training data, regardless of the task.
>
> Naturally, whether this can generalize to test data depends on the model’s (mis)match to the true data-generating process.
>
> > 3. Can we view DAML as “finding the global optima” (in terms of BPR) that “has the highest log likelihood”?
>
> Yes, we agree with this view. There can be different models $\phi$ that have equivalent rankings r and thus equivalent BPR value. We can view DAML with a high BPR constraint as seeking $\phi$ that have both high BPR and high log likelihood.
>
> > Do the results remain similar if minibatching is used? I understand that for the datasets used minibatching may not be necessary, but for larger datasets one would certainly like to use it.
>
> Minibatch gradient descent to minimize the BPR only loss in Eq 9 or the DAML objective in Eq 13 appears straightforward in both cases. In each case, the overall “whole dataset” loss is a sum over the loss at each timestep indexed by t. Taking a random subset of timesteps as a minibatch and computing the gradient of that minibatch should be an unbiased estimate of the gradient of the whole-dataset loss.
>
> We have not tested minibatching in our experiments, as it wasn’t necessary and would introduce extra noise to our gradient-based learning that is already noisy (due to score function trick and perturbations). Tuning learning rates and batch sizes carefully would be important, but if done well, minibatching seems feasible.

---

### Official Review · Reviewer_wWV7 · 2025-03-14

**Overall Recommendation:** 4

**Summary:**

This paper studies measures of best possible reach (BPR) to select the best subset of interventions. They analyze different measures based on a probabilistic model, as well as different way to train these probabilistic models from historical data. They also propose new methods for training, including a decision-aware maximum likelihood solution which empirically achieves the best trade-off between BPR and test log-likelihood. They evaluated their methods on synthetic data and real overdose forecasting data.

**Claims And Evidence:**

The claims made in the submission are clear, and convincingly supported by experiments.

**Essential References Not Discussed:**

N/A

**Experimental Designs Or Analyses:**

The experimental designs and analyses are sound.

**Methods And Evaluation Criteria:**

The methods (ranking and training) make sense for the application at hand.

**Other Comments Or Suggestions:**

One suggestion I could make is to expand on the minimization of $J^{BPR}$ when the model is misspecified (lines 255-260), and why the forecasts have questionable utility. This is seems to be related to the observation in Figure 2 (left), but this is not obvious to a reader less familiar with this field.

Another suggestion is to highlight the fact that DAML can also integrate prior knowledge if available (which I imagine might be the case in practice?). This is similar to how we can move from MLE to MAP (as mentioned by the authors), but is crucially different from the BPR loss where there is a priori no obvious way to incorporate this type of knowledge.

**Other Strengths And Weaknesses:**

Despite not being familiar with this literature, this paper seems to be a strong practical contribution for ICML. The pedagogical approach of the different methods for ranking and training is greatly appreciated. The rigor shown in the experimental results is valuable, and the results convincing on the advantages of DAML as a trade-off between test log-likelihood and BPR.

The authors are also very explicit in the limitations of their work, and possible (realistic) directions for future work.

**Questions For Authors:**

N/A

**Relation To Broader Scientific Literature:**

N/A

**Theoretical Claims:**

N/A

---

> ### Author Rebuttal · Authors · 2025-04-01
>
> We thank the reviewer for their time and helpful feedback. We are glad to hear the overall story of our manuscript made sense to you.
>
> We offer a few responses to the questions you raised below:
>
> > One suggestion I could make is to expand on the minimization of JBPR when the model is misspecified (lines 255-260), and why the forecasts [for BPR only] have questionable utility.
>
> Thanks for this idea, we will revise to improve the clarity of this point.
>
>
> Essentially, the issue is that when only training to maximize BPR, all that matters is the ranking of each site in the r vector. Nothing about the implied distribution over y values is forced to match the true distribution of y, beyond just getting the ranking order correct so the predicted set of top K sites aligns with the true top K
> Consider the seven sites in the toy example in Fig 2. Imagine two possible r vectors, each defined as a per-site mean of a distinct model for $p(y_1, … y_7)$
>
> ```
> site  1  2  3  4  5  6   7
>  rA: 10 20 30 40 50 60 100
>  rB:  1  2  3  4  5  6   7
> ```
> Both rA and rB would have the same BPR, reaching a perfect 1.0, because they rank the top 5 sites (#3 - #7) correctly. However, only model A is at all near the true per-site means indicated in the true-distribution histograms in Fig. 2. Model B has very questionable utility for forecasting y values, because it says the per-site mean of site 7 is 7, which far lower than the actual mean of site 7 (around 100).
>
> > Another suggestion is to highlight the fact that DAML can also integrate prior knowledge if available (which I imagine might be the case in practice?). This is similar to how we can move from MLE to MAP (as mentioned by the authors), but is crucially different from the BPR loss where there is a priori no obvious way to incorporate this type of knowledge.
>
> Thanks, we will revise to highlight the ability to integrate prior knowledge via the model. Indeed, in the real applications, we do incorporate a prior on the random effect coefficients (see App. E), as recommended by past work on this negative binomial regression model in conventional settings.

---

### Official Review · Reviewer_ZTv9 · 2025-03-18

**Overall Recommendation:** 3

**Summary:**

The paper proposes an approach for learning a potentially misspecified spatialtemporal probabilistic model for decision-making settings. They specifically focus on the decision problem optimizing best possible reach (BPR) which closely corresponds to the ranking top-K items problem. Their approach consists of first proposing a metric similar to BPR in order to rank the items. They then show how to compute the metric given a probabilistic model. Given the metrics they then show how to compute the BPR loss and how to optimize the loss relative to the probabilistic model. Putting it all together, they propose a loss that blends decision loss (BPR loss) with traditional ML loss (negative log likelihood). This approach allows users to trade off interpretability (ML loss) with decision performance (BPR loss) via a tuning parameter.
 Finally, they test their approach on three real-world datasets to highlight the benefits of their approach.

**Claims And Evidence:**

In general, this paper provides examples and empirical evidence to support their proposed approach. The paper does not explicitly provide any theoretical justification for their approach, but do cite other papers when deriving key quantities like the gradient for the ratio estimator. The paper does not have any obvious problematic claims.

**Essential References Not Discussed:**

The paper does not really cite decision-aware literature which can be is well summarized in [1],[2], and [3]. The last paper especially provides computational approaches in a python package for computationally solving the decision-aware BPR problem.

They also do not cite applications of decision-aware approaches such as [4].

Finally, the authors may also benefit from comparing their approach to [5] which also proposes a decision-aware approach and a case study that closely resembles the BPR problem.

[1] Mandi, Jayanta, et al. "Decision-focused learning: Foundations, state of the art, benchmark and future opportunities." Journal of Artificial Intelligence Research 80 (2024): 1623-1701.
[2] Tang, Bo, and Elias B. Khalil. "Pyepo: A pytorch-based end-to-end predict-then-optimize library for linear and integer programming." Mathematical Programming Computation 16.3 (2024): 297-335.
[3] Sadana, Utsav, et al. "A survey of contextual optimization methods for decision-making under uncertainty." European Journal of Operational Research 320.2 (2025): 271-289.
[4] Chung, Tsai-Hsuan, et al. "Decision-aware learning for optimizing health supply chains." arXiv preprint arXiv:2211.08507 (2022).
[5] Gupta, Vishal, Michael Huang, and Paat Rusmevichientong. "Decision-aware denoising." Available at SSRN 4714305 (2024).

**Experimental Designs Or Analyses:**

I reviewed the synthetic data experiment, the opioid-related overdose forecasting experiment, and the endangered bird forecasting experiment. The synthetic data experiment provided a simple example highlighting the potential pitfalls of only using traditional ML loss and only using BPR loss. The real-world datasets seem to closely follow the experimental set-up of previous works.

**Methods And Evaluation Criteria:**

The high level idea makes sense as the method provides users a heuristic approach for trading off decision quality and prediction quality. It makes sense to construct a surrogate for decision-loss in order to incorporate it into learning the underlying probabilistic model. The datasets provided by the authors also fit the setting of optimizing best possible reach and seem to have been used in previous papers for the same application.

**Other Comments Or Suggestions:**

1. It may help the paper to better justify the choices in Section 3. The paper seems to present two methods of ranking, but in the experiments only present the results of the second approach. Section 3 seems to be the most novel component of the decision-aware approach, so explaining the modeling choices would help highlight the differences between existing decision-aware learning approaches.

2. The paper's title mentions spatialtemporal decision-making settings, however, aside from Eq. (15) there seems to be little consideration of the spatial or temporal aspects of the data. Highlighting components of the paper that leverage the structure of space and time would help differentiate it from existing work.

**Other Strengths And Weaknesses:**

Strengths
1. The paper provides compelling real-world datasets and experiments to highlight the benefits their approach.
2. The paper provides good motivation for blending traditional ML loss with decision-aware loss.

Weaknesses
1) The paper provides limited theoretical or practical justification for their proposed BPR loss. My main concern is that the loss proposed by the authors seem to equally weight the BPR loss of each time period $t$. Thus, they are maximizing the average historic BPR. However, since they consider temporal elements, a more sensible decision loss to optimize would be the BPR loss of time period T+1. Optimizing over average historic BPR instead of the BPR of time period T+1 may reduce the decision quality of the learned probabilistic models. [1] highlights how to estimate the decision loss at time period T+1 which may be an interesting benchmark to consider.

2) The paper does not consider other more popular decision-aware approaches. It can be shown that the BPR problem could be formulated as a 0-1 knapsack problem since the denominator term $\bf{y} \cdot \text{TopKMask}(\bf{y},K)$ is a constant (does not depend on $r$) and $\text{TopKMask}(\bf{r},K)$ selects the $K$ largest elements in $\bf{r}$. This should allow the authors to optimize the BPR loss with decision-aware approaches found in the python package PyEPO [2]. The approaches in the package should be compatible with the BPR problem since the package only requires users providing the gradients of $r^{*}(\phi)$ and the 0-1 knapsack formulation. As shown in [3], the choice of the approach can affect the resulting decision quality of the learned prediction model.

3) The paper's novelty seems limited. The paper's contributions can be grouped into i) computation and ii) formulation. From a computational perspective, the challenges can be addressed by existing work as discussed in the previous point 2) and the authors do not compare their approach to these existing approaches. From a formulation perspective, the paper's constrained optimization problem reduces to an unconstrained optimization problem that takes a weighted combination of the ML loss and decision loss. This has been previously proposed in [4]. Moreover, the construction of the decision loss is not well justified as highlighted in point 1).
_____
[1] Gupta, Vishal, Michael Huang, and Paat Rusmevichientong. "Decision-aware denoising." Available at SSRN 4714305 (2024).
[2] Tang, Bo, and Elias B. Khalil. "Pyepo: A pytorch-based end-to-end predict-then-optimize library for linear and integer programming." Mathematical Programming Computation 16.3 (2024): 297-335.
[3] Huang, Michael, and Vishal Gupta. "Decision-focused learning with directional gradients." Advances in Neural Information Processing Systems 37 (2024): 79194-79220.
[4] Kao Yh, Roy B, Yan X (2009) Directed regression. In: Bengio Y, Schuurmans D, Lafferty J, Williams C, Culotta A (eds) Advances in Neural Information Processing Systems, Curran Associates, Inc., vol 22

**Questions For Authors:**

Below summarizes some of the questions mentioned in the rest of the review:
1. Is top K not just a 0-1 knapsack problem of size K? In that case why can you not reformulate the problem as a linear program? You have the gradient of the estimator in (10). Methods found in the PyEPO package (https://github.com/khalil-research/PyEPO) like SPO+, DBB, and PG Loss are compatible. SPO+ is a convex surrogate while PG Loss works well for misspecified models.

2. Why is your proposed loss (9) a good measure of decision loss? Is it some unbiased estimate of the decision loss of time T+1?

3. Why does using only BPR loss provide worse Test BPR for the MA opioid-related overdose forecasting? This seems different than the other two Pareto frontiers shown in Figure 3.

4. In your results, can you show that you can control the BPR with $\epsilon$?

5. Can you apply the score function trick to the quantity $\frac{y_i}{\bf{y} \cdot \text{TopKMask}(\bf{y},K)}$? Using this quantity instead of the ratio estimator would be a more direct way to rank the items.

**Relation To Broader Scientific Literature:**

The key contribution of the paper is showing how to solve the best possible reach (BPR) problem in a decision-aware way. This builds off existing decision-focused and end-to-end learning literature [1][2].  They use compelling real-world data sets and show that decision-aware methods can be effective. The paper outlines the challenges of solving the decision-aware problem and highlights solutions that leverage the score function trick [3] and perturbed optimization [4].

In the decision-aware literature, this work most closely resembles [5] which also focuses on applying decision-aware approaches to concrete real world problem.

-----
[1] Mandi, Jayanta, et al. "Decision-focused learning: Foundations, state of the art, benchmark and future opportunities." Journal of Artificial Intelligence Research 80 (2024): 1623-1701.
[2] Tang, Bo, and Elias B. Khalil. "Pyepo: A pytorch-based end-to-end predict-then-optimize library for linear and integer programming." Mathematical Programming Computation 16.3 (2024): 297-335.
[3] Mohamed, S., Rosca, M., Figurnov, M., and Mnih, A. Monte carlo gradient estimation in machine learning. Journal of Machine Learning Research, 21(132), 2020.
[4] Berthet, Q., Blondel, M., Teboul, O., Cuturi, M., Vert, J.-P., and Bach, F. Learning with Differentiable Perturbed Op- timizers. In Advances in Neural Information Processing Systems (NeurIPS), 2020.
[5] Chung, Tsai-Hsuan, et al. "Decision-aware learning for optimizing health supply chains." arXiv preprint arXiv:2211.08507 (2022).

**Theoretical Claims:**

The paper makes no theoretical claims.

---

> ### Author Rebuttal · Authors · 2025-04-01
>
> We thank ZTv9 for their thoughtful review, especially in introducing related work and PyEPO. We offer brief replies to key points below. We will revise to address all points raised.
>
> ## Essential References
>
> Thanks for several useful references. Please see “Essential References (Common Issue)” in Response to Reviewer His4 for our revision plan.
> ## W1 & Q2: Does decision loss optimize for time T+1?
> > Optimizing over average historic BPR instead of the BPR of time period T+1 may reduce the decision quality
>
> Empirically, we find decisions at heldout times are good (Fig 3). Revision will discuss justification below, and mention out-of-sample bounds by Gupta et al (SSRN ‘24).
>
> We assume the true data-generating distribution of outcomes at time t given the recent past, $p(y_t | y_{t-W:t-1})$, is unchanged across train and test time periods. We will make this explicit in revision. Many forecasting methods use this assumption and use average loss to extrapolate to the future. We could weight recent timesteps more.
>
> As further protection, we already use a “leave future out” experimental design: e.g. for MA overdose task, we train on 2011-18, tune/validate on 2019, and test on 2020-21.
>
> ## W2 & Q1: 0-1 Knapsack for BPR / PyEPO baseline
>
> Thanks for suggesting the 0-1 knapsack formulation. The problem of “how to rank” for BPR given outcome y can reduce to 0-1 knapsack where each site has weight 1 and the budget constraint allows K of S sites.
>
> As suggested, we tried both SPO+ loss and PG loss from PyEPO on the 3 datasets in our Fig. 3. Both losses are direct competitors to our BPR-only (Sec 4.2), as they try only to improve top-K decisions and do not account for likelihood.
>
> Results are in [revised Fig 3](https://anonymous.4open.science/r/DecisionAwareMaximumLikelihood/Figure3Update.pdf) in our anonymous code repo.
>
> We find SPO+ and PG do not advance any panel’s Pareto frontier compared to our methods. In terms of top-K decisions, each method can sensibly beat NLL-only but not our DAML or BPR-only. PG loss gets higher BPR than NLL-only on Cook, SPO+ does the same for MA. However, these objectives ignore likelihood, so they naturally produce models with low likelihood.
>
> ## W3: Novelty
>
> Our DAML approach in Sec 4.3 is not wholly new in combining classic objectives with decision-aware losses. We will revise to cite Kao et al. (NeurIPS 2009)’s convex-combination of these two loss types, albeit for directed regression rather than our top-K where-to-intervene tasks. Compared to that work, DAML is distinct in its *constrained* approach: as soon as the desired BPR is achieved, training focuses only on likelihood.
>
> Another novel aspect of our work is the ratio estimator for the how to rank problem (Sec 3). While previous works used BPR as a metric, they defaulted to the per-site mean, unaware that it may be suboptimal (see App. B for demo of suboptimality).
>
> A final novel aspect is our analysis of 3 real where-to-intervene datasets, covering whooping-crane and opioid-overdose forecasts, each with 1000+ sites. These tasks are not yet in any decision-aware literature to our knowledge, and use models much bigger than in Gupta et al. (SSRN ‘24)
>
> ## Q3: Why does using only BPR loss provide worse Test BPR for the MA overdose forecasting?
>
> Yes, the left panel of Fig 3 shows a different relative ranking of BPR-only vs. our DAML method. Optimizing BPR is prone to local optima and difficult loss landscapes. The extra NLL loss in DAML appears to find better BPR solutions sometimes.
>
> ## Q4: Can you control BPR with epsilon?
>
> Yes, see Pareto frontier in our Fig 2. On this toy task, as we raise epsilon from low to 0.86 to 1.0, the BPR on test data from the true generating process follows the expected trajectory. On real data (Fig. 3), the epsilon at training has a looser relationship with BPR on a limited test set due to mismatched assumptions.
>
> ## Q5: Can you use a ranking with top-K in denominator?
>
> The proposed estimator is:
> $$
> r(\phi) = \mathbb{E}[ \frac{ y }{ y \cdot {TopKMask}(y, K) } ]
> $$
> This differs from our ratio estimator (Eq 7) by including the top-K binary mask in the denominator, instead of an all-ones vector.
>
> In expectation, our ratio estimator will have the same ranking of the S sites as this proposal. Thus, this estimator will produce the same top-K decisions as our ratio estimator. We prefer our ratio estimator because it is simpler and faster (avoids top-K for each sample y).
>
> We will update the Appendix to discuss our reasons for preferring our ratio estimator.
>
> ## Other
>
> > better justify the choices in Sec. 3
>
> See App. B for detailed examples where given the same parameters, our ratio estimator can deliver BPR 2x higher than the per-site mean. We will revise to clarify further.
>
> > consideration of the spatial or temporal aspects
>
> All models in Fig. 3 use as a feature for site s the gravity of its spatial neighborhood, that is, a recent average of events in sites spatially near to site s.

---

### Official Review · Reviewer_7cWK · 2025-03-22

**Overall Recommendation:** 3

**Summary:**

The paper tackles two main issues related with a metric called Best Possible Reach (BPR): (1) the ranking problem, basically how to rank sites numerically to select the top K for intervention, based on a probabilistic method, to solve this, the paper works on a tighter bound on BPR and utilizes the ratio estimator. (2). the training problem, basically how to optimize the model's parameter to maximize BPR performance. The paper tackles the difficult problem of training models to directly optimize the BPR metric, which involves a discrete top-K selection that leads to zero gradients. It uses the perturbed optimizers and the score function trick to estimate gradients and enable training. Experiments are done on synthetic data and two real world datasets: opioid overdose mitigation and endangered bird monitoring.

**Claims And Evidence:**

Some claims made by the paper:
- Ranking via the per-site mean is suboptimal for BPR. Section 3.1 shows the per-site mean minimize expected loss on a simplistic upper bound on BPR.
- The effectiveness of the proposed ratio estimator. Intuitively, it works under the sparse vector setting, and does provide a tighter upper bound. This is also empirically proved in Fig 2.
- DAML helps navigating the pareto frontier, both experiments on synthetic data and real-world datasets consistently verify this empirically.

**Essential References Not Discussed:**

NA

**Experimental Designs Or Analyses:**

See Methods And Evaluation Criteria.

**Methods And Evaluation Criteria:**

- In the synthetic experiments, the approach is evaluated only on the very specific models: Gaussian mixtures and negative binomial mixed effects, which might seem too simple for realistic uses cases? Are these still the most common used models? Can the authors comment more on the model's performance on a more complex model families?
- I am concerned about the scalability of the proposed method, it seems evaluated on a small scale of parameters $T, S$ and $K$. Could the author comment more on the possible approximate algorithms people might be able to use when face more real-world large scale problems?
- The approach comes with a pre-defined $K$. Typically how people find this $K$ value? How does the method perform robustly across a wide range of $K$'s? Can we add more empirical results on this?
 - Both the hyper-parameter $\sigma$ and $\lambda$ are key hyper-parameters needs to be tuned. Could the authors provide some rule of thumb (or automatic selection methods) in selecting them under different use cases?
- Though the paper explicitly discusses the benefits of their approach under model misspecification, the synthetic experiment uses a relatively simple form of misspecification. It would be interesting to see how the methods perform under more severe or diverse types of misspecification.

**Other Comments Or Suggestions:**

See above.

**Other Strengths And Weaknesses:**

Strength:
- The paper is well-written and solves some key problems with BPR metrics, in terms of ranking and optimization.
- The authors propose a "ratio estimator" for ranking sites based on a tighter bound of the BPR metric. It moves beyond the suboptimal per-site mean ranking and is theoretically justified. Empirically, this method significantly outperforms the per-site mean.
- The proposal of DAML provides a nice way to balance the goals of achieving high BPR for decision-making and maintaining good likelihood for overall forecast accuracy.

Weakness:
- The scalability of the method.
- The effectiveness of the method moving beyond the two families of the models examined in the paper.
- The robustness regarding to the pre-determined K value.

**Questions For Authors:**

See Methods And Evaluation Criteria.

**Relation To Broader Scientific Literature:**

Might be related with spatial-temporal forecasting topics; method-wise, might related with BPR, direct loss minimization, and score function estimator.

**Theoretical Claims:**

I roughly looked at the ones in the main paper, and does not identify any obvious issue.

---

> ### Author Rebuttal · Authors · 2025-04-01
>
> We thank the reviewer for their thoughtful feedback. We are glad that they thought our DAML method “provides a nice way to balance the goals of achieving high BPR for decision-making and maintaining good likelihood for overall forecast accuracy.”
> > the approach is evaluated only on …Gaussian mixtures and negative binomial mixed effects, which might seem too simple for realistic use cases? Are these still the most common used models?
>
> For our real applications, we selected the negative binomial mixed effects regression with spatially-lagged features (Sec. 5.2), specifically because it had competitive heldout performance in a recent evaluation of methods for opioid-overdose forecasting published in 2024 by Heuton et al. [A]. In that study, the negative binomial model with conventional training (not the decision-aware training in our submission) tied or beat an attention-based neural network as well as a boosted ensembles of trees model and a Gaussian process model in terms of test-set BPR on two different datasets.
>
> These spatiotemporal forecasting tasks are overall quite difficult; there is not often a very strong signal for predicting observed counts of overdose deaths or bird sightings given the limited available features. In such cases, somewhat simpler models tend to work well, even when substantial effort is put into tuning hyperparameters to avoid overfitting with complex tree ensembles or neural nets.
>
> **New Results with GNNs**. We have also added a [New Table](https://anonymous.4open.science/r/DecisionAwareMaximumLikelihood/NewTable.pdf) as a standalone PDF in our anonymous code repo. This compares our methods to a recent graph neural networks for spatiotemporal forecasting (from Xie et al. ‘24 [B]) on the Crane and Cook County tasks, in terms of BPR, RMSE, and MAE. This GNN is trained to minimize squared error, and thus can do best on RMSE but is inferior to our DAML for top-K decisions as measured by BPR.
>
> We do hope to investigate a broader class of probabilistic models directly trained by our DAML in future work.
>
> [A] Heuton et al. Spatiotemporal forecasting of opioid-related fatal overdoses. Amer. J. of Epidemiology, 2024.
>
> [B] Xie et al. EpiGNN: Exploring Spatial Transmission with Graph Neural Network for Regional Epidemic Forecasting. ECML PKDD ‘22.
>
> > concerned about the scalability of the proposed method, it seems evaluated on a small scale of parameters T, S, and K. . Could the author comment more on the possible approximate algorithms
>
> We agree that scalability is an important practical consideration. Our current code can scale to number of timesteps T in the dozens, budget K in the hundreds, number of sites S in the thousands, and number of parameters P in the thousands. We have found this sufficient for the practical problems in our paper.
>
> Our algorithm can further be scaled up to larger problems by processing one time record (out of T) at a time, and also by computing gradients with respect to one parameter (out of P) at a time. We have prototyped code to do this already in the last few weeks.
>
> With further implementation effort, our methods could be easily adapted to process minibatches of timesteps, to scale even further.
>
> > The approach comes with a pre-defined K. Typically how people find this value? … does the method perform robustly across a wide range?
>
> Typically, the practical intervention budget of the stakeholders determines K. For example, in the whooping crane monitoring application shown in Fig. 1, if the monitoring agency can only afford 10 new cameras, then we would set K=10.
>
> We will try to do some further experiments on different K values for one of the applied datasets in the next few weeks. If accepted, we promise to include such experiments in the appendix for the camera-ready deadline.
>
> > Both sigma and lambda are key hyper-parameters needs to be tuned. Could the authors provide some rule of thumb?
>
> We will revise the paper to provide this information. To complete our experiments in Fig. 3, we select sigma from a grid of values between 0.001 and 0.1 based on validation-set loss. We observed the experiments to be insensitive to lambda, and it was fixed at 30 to make the BPR and likelihood components of the loss similar in magnitude during early training.
>
> > the synthetic experiment uses a relatively simple form of misspecification. It would be interesting to see how methods perform under more severe or diverse types of misspecification.
>
> We agree that exploring more types of misspecification would be interesting. However, with limited time in this rebuttal period, we elect to leave this to future work. We will revise to acknowledge this limitation in our Discussion section.

---

### Decision · Program_Chairs · 2025-05-01

**Decision:**

Accept (poster)

**Comment:**

I concur with the overall positive sentiment from the reviewers and agree the submission would make a nice addition to the conference. The reviewers gave some great feedback that should be carefully incorporated into the final camera ready. The most important is making very clear the connections to contextual stochastic/linear optimization (using the reduction explained by reviewer ZTv9) and referencing the relevant papers in that literature (for reference a comprehensive list would be these arxiv ids 1402.5481 1710.08005 1905.11488 2008.07473 2011.03030 2402.03256 2405.16564) as well as discussing the other decision-aware learning papers mentioned.